

# Single field-of-view sounder atmospheric product retrieval algorithm: establishing radiometric consistency for hyper-spectral sounder retrievals

Wan Wu[1], Xu Liu[1], Liqiao Lei[2], Xiaozhen Xiong[1], Qiguang Yang[2], Qing Yue[3], Daniel K. Zhou[1], Allen M. Larar[1]

[1]NASA Langley Research Center, Hampton, VA 23682, USA
[2]Sceince Systems and Applications, Inc., Hampton, VA23666, USA
[3]Jet Propulsion Laboratory, California Institute of Technology, Pasadena, CA 91109, USA

**Abstract.** The Single Field-of-view (SFOV) Sounder Atmospheric Product (SiFSAP) retrieval algorithm has been developed to address the need to retrieve high spatial resolution atmospheric data products from hyper-spectral sounders and ensure the radiometric consistency between the retrieved properties and measured spectral radiances. It is based on an integrated optimal estimation inversion scheme that processes data from the satellite based synergistic microwave (MW) and infrared (IR) spectral measurements from advanced sounders. The retrieval system utilizes the principal component radiative transfer model (PCRTM) which performs radiative transfer calculations monochromatically and includes accurate cloud scattering simulations. SiFSAP includes temperature, water vapor, surface skin temperature and emissivity, cloud height and microphysical properties, and concentrations of essential trace gases for each SFOV at a native instrument spatial resolution. Error estimations are provided based on a rigorous analysis for uncertainty propagation from the Top-Of-Atmosphere (TOA) spectral radiances to the retrieved geophysical properties. As a comparison, the spatial resolution for the traditional hyper-spectral sounder retrieval products is much coarser than the native resolution of the instruments due to common use of the 'cloud clearing' technique to compensate for the lack of cloud scattering simulation in the forward model. The degraded spatial resolution in traditional cloud-clearing sounder retrieval products limits their applications for capturing meteorological or climate signals at finer spatial scales. Moreover, rigorous uncertainty propagation estimation needed for long-term climate trend studies cannot be given due to the lack of direct radiative transfer relationships between the observed TOA radiances and the retrieved geophysical properties. With advantages of the higher spatial resolution, the simultaneous retrieval of atmospheric, cloud, and surface properties using all available spectral information, and the establishment of 'radiance closure' in the sounder spectral measurements, the SiFSAP provides additional information needed for various weather and climate studies and applications using sounding observations. This paper gives an overview of SiFSAP retrieval algorithm, and assessment of SiFSAP atmospheric temperature, water vapor, clouds, and surface products derived from the Cross-track Infrared Sounder (CrIS) and Advanced Technology Microwave Sounder (ATMS) data.



## 1. Introduction

Since the launch of the first space-borne hyper-spectral infrared (IR) sounder, the Atmospheric Infrared Sounder (AIRS), the value of spectrally-resolved IR measurements for weather forecasting (LeMarshall et al., 2006, Chahine et al., 2006, Jones et al., 2012), environmental monitoring (Chahine et al., 2008, Warner et al., 2017, Ribeiro et al., 2018, Nalli et al., 2020), and the study of climate forcing and feedbacks (Gettelman and Fu, 2018, McCoy et al., 2019, Liu et al., 2018) have been widely recognized. Hyper-spectral IR sounders like AIRS, Cross-track Infrared Sounder (CrIS), and the Infrared Atmospheric Sounding Interferometer (IASI) measure the outgoing longwave radiation using thousands of spectral channels. They are designed to achieve high vertical resolution sounding of atmospheric temperature and humidity profiles, to provide spectral information for the retrieval of cloud phase, height, and microphysical properties, and to capture spectral signatures of key trace gases. Multiple operational retrieval algorithms have been developed to generate Level-2 products of geophysical properties from Level-1 spectral radiance data. Examples of operational algorithms include the regression based algorithms such as the dual-regression algorithm (Smith et al., 2012), the physical algorithms such as the Climate Heritage AIRS Retrieval Technique (CHART, Susskind et al., 2017), the Community Long-term Infrared Microwave Combined Atmospheric Product System (CLIMCAPS, Smith and Barnet, 2019), and the NOAA-Unique Combined Atmospheric Processing System (NUCAPS, Barnet, 2021), and the hybrid algorithms that perform physical retrieval for clear sky cases and regression for cloudy sky retrievals, e.g., the Level 2 IASI Product Processing Facility (PPF, August et al., 2012).

There are ongoing efforts to exploit the use of hyper-spectral sounder measurements for new applications with requirements that have yet to be met by the operational sounder products mentioned above. The limits on the applications of these Level-2 products come from two perspectives: the degradation of spatial resolution as compared with the native resolution of the instruments and the lack of radiative closure between the retrieved geophysical properties and the TOA spectral measurements. Specifically, operational Level-2 data products from physical retrieval schemes including CHART (Susskind et al., 2017), CLIMCAPS (Smith and Barnet, 2019), NUCAPS (Barnet et al., 2021) uses 3 × 3 IR sounder field of views (FOVs) (along track × across track) to construct a 'cloud-cleared' single spectrum that is reregistered with one sounder field of regard (FOR). As a result, the spatial resolution of the Level-2 properties is reduced by a factor of 3, i.e. 9 times less retrieved data. The degradation of spatial resolution limits the applications of these operational sounder products in various studies, such as tracing the source and propagation of gravity waves (Sato et al., 2016, Ern et al., 2017, Perrett et al., 2021), studying the impact of convection on Planetary Boundary Layer (PBL) thermodynamics (Elsaesser et al., 2019), and constructing vertical profiles of winds using temperature, humidity, and ozone profiles, etc. Existing SFOV products obtained using non-physical algorithms (e.g. dual-regression and IASI PPF for cloudy-sky cases) tend to be more error prone due to the lack of radiometric closure with the observed radiance spectra. Therefore, there is a growing demand to develop SFOV physical retrieval schemes for hyper-spectral sounder data applications. The SFOV retrieval methodology was first introduced to process airborne campaign data from the National Airborne Sounder Testbed-Interferometer (NAST-I) onboard the NASA suborbital ER-2 aircraft (Cousins and Smith, 1999). Atmospheric profiles together with cloud microphysical properties and surface properties can be



retrieved under all sky conditions (Zhou et al., 2005, 2007, Liu et al., 2007). The study of using SFOV methodology for satellite

based hyper-spectral IR sounder measurements has been eventually carried out (Liu et al., 2009, Zhou et al., 2009, Wu et al., 2017, Irion et al., 2018, DeSouza-Machado et al., 2018). As the SFOV methodology matures, its operational application for hyper-spectral IR sounder missions has become very promising.

Establishing radiative closure, i.e. the radiometric consistency of the TOA spectra from radiative forward modelling using retrieved geophysical properties with respect to the observations, is critical to studies of climate trends and anomalies. The

accuracy of climate trends derived from hyper-spectral IR observations depends on the radiometric accuracy of the measurements and a rigorously defined inverse relationship that links the measurements to the climate variables of interest (Liu et al., 2017). Non-physical retrieval schemes do not establish radiative closure by their nature. The closure in physical retrieval schemes including CHART, CLIMCAPS, NUCAPS and the hybrid IASI PPF can only be established for clear sky observations which just account for a small percentage of the global measurements. Without including cloud scattering in the

forward simulations, the impact of radiometric uncertainty on the retrieved climate variables cannot be directly characterized. Estimation for radiometric errors and/or discontinuities and the corresponding impact on climate variables retrieved is critical for the construction of long-term climate anomalies and/or trends data record. From this perspective, a physical retrieval algorithm that establishes radiative closure by simulating cloud scattering in the radiative transfer process is more suitable to produce accurate, long-term climate data records.

The SiFSAP retrieval algorithm has been developed to supplement other operational products by improving the spatial resolution and establishing the radiative closure. The principal component radiative transfer model (PCRTM, Liu et al., 2006) is used for the forward simulation of the hyper-spectral IR sounder spectra in the SiFSAP system.  PCRTM uses empirical orthogonal functions (EOFs) to compress the spectral information so that the complete spectrum of hyper-spectral sounder measurements from the full set of channels can be efficiently used. It facilitates an accurate multiple cloud scattering

calculation by using lookup tables constructed via 32-stream Discrete Ordinates Radiative Transfer (DISORT) simulations (Stamnes et al., 1988). The SiFSAP algorithm simultaneously retrieves profiles of temperature, moisture, and trace gases of interest, surface properties, and cloud parameters including visual optical depth, particle size, phase, and height. The solution is obtained by fitting the TOA spectrum for each single FOV observation via an iterative minimization process following the optimal estimation method (OEM) (Liu et al., 2007, Liu et al., 2009, Wu et al., 2017). Compared to other retrieval algorithms,

the radiative relationships between the retrieved geophysical properties and the measured TOA radiances are rigorously and consistently defined for both clear and cloudy sky conditions in the SiFSAP scheme, therefore the radiative closure is established.

Leroy et al. (2018) found that erroneous priors used in AIRS retrievals introduce systematic biases in the anomalies of stratospheric temperature over Antarctica. Using stringent *a priori* reduces the uncertainty in individual retrievals while making

the results more prone to systematic errors given faulty *a priori*. The SiFSAP algorithm uses the climatology-based *a priori* for two important considerations: 1) uncertainty of individual measurements is less a concern as compared with systematic bias in long-term climate variability studies, and 2) the climatological *a priori* constraint constructed from globally distributed



data maximizes the information determined from the radiances and minimizes the impact from *a priori* errors. The final solutions of SiFSAP usually deviate significantly from the first guess, i.e., the global mean of the climatological data, used in

the retrieval (see Figure 1). This is very different from CHART and CLIMCAPS that constrain the results around the first guess, e.g., the deviation of retrieved temperature from its first guess value is less than 1 K (Wang et al., 2020, Yue et al., 2020). Avoiding the use of auxiliary data products as prerequisites enables the SiFSAP system to meet the primary latency requirements imposed on near real-time algorithms. Therefore, the SiFSAP algorithm is suitable for both climate and weather applications.

This paper gives a detailed introduction on the data content, the data processing scheme, and the physical retrieval methodology of SiFSAP. Validation of retrieval performance for SiFSAP key products is also presented.

## 2. Overview of the data content and the retrieval system

The SiFSAP system is designed to process data from major hyper-spectral sounders including AIRS, IASI and CrIS. It can be easily extended to process future sounders like IASI-Next Generation once the requisite PCRTM module is correspondingly

updated. The system depends upon PCRTM's capability to simulate various hyper-spectral sounder measurements using a common forward model module. Previous forward model comparison studies have validated PCRTM's capability in simulating hyper-spectral sounder measurements with a high degree of radiometric fidelity (Aumann et al., 2018). Figure 2 demonstrates the use of SiFSAP to simulate sample TOA spectral radiances measured by three major hyper-spectral sounder instruments: IASI, CrIS, and AIRS. Benefiting from its modular design, the SiFSAP system is capable of using the collocated

microwave (MW) observations to supplement IR retrievals under thick cloud conditions. Currently the SiFSAP system provides three retrieval schemes to meet different application needs based on the observation data availability: IR-only, MW-only, and IR+MW retrievals. The MW retrieval unit uses the Community Radiative Transfer Model (CRTM) to simulate the measurements by major MW sounders including the Advanced Technology Microwave Sounder (ATMS), the Advanced Microwave Sounding Unit (AMSU-A), and the Microwave Humidity Sounder (MHS). Details about CRTM can be found in

CRTM user's guide (Han *et al.,* 2005) and its general introduction paper (Liu *et al.,* 2012). SiFSAP products are generated using the synergistic IR+MW retrievals with IR and MW spectra being fitted simultaneously during the combined retrieval process. SiFSAP products include temperature, water vapor, and trace gas profiles at 98 pressure levels, surface skin temperature, IR surface emissivity at native mono-frequency bins defined by the PCRTM, MW surface emissivity for all MW sounder channels, effective cloud top pressure, cloud optical depth (at 550 nm), cloud particle size, and cloud liquid water

content. Table 1 lists the major geophysical properties included in SiFSAP.

Figure 3 illustrates the flow diagram of the SiFSAP system. The SiFSAP processing starts with a pre-processor that loads the Level-1B data of the IR and MW sounders and the surface pressure values from the National Centers for Environmental Prediction (NCEP) Global Forecast System (GFS) model fields. The MW sounder data are spatially resampled to overlap with IR sounder observations of single FOVs via nearest neighbour gridding. The GFS surface pressure data are interpolated in

time and space to the IR sounder footprints. At the synergetic data processing stage, the SiFSAP system includes three modular



units: the initialization unit, the synergetic retrieval unit, and the post-processing unit. The initialization unit loads static databases including PCRTM and CRTM forward model parameters, lookup tables (LUTs), climatological background fields, *a priori* covariance matrices, measurement uncertainties covariance matrices, and pre-trained spectral bias correction coefficients. The synergetic retrieval unit includes a two-step process: MW only retrieval followed by IR + MW combined retrieval. The temperature, water vapor, and surface skin temperature from the first step MW-only retrieval, once passed the MW radiance convergence test, are used as the first guess for the combined IR+MW retrieval. If the MW retrieval does not pass the first step convergence test, the climatological first guess is used for the combined retrieval. If the geophysical properties retrieved for one FOV pass the quality control (QC), they are used as the first guess values for the next FOV. Again, the climatological first guess will be used for the next FOV if the retrieval of the current FOV does not converge. The MW and the IR+MW combined retrieval results are passed to the QC and post-processing unit where QC flags are assigned, auxiliary data such as the tropopause height and the surface temperature are derived based on the retrieved atmospheric parameters, and the results are written to output files.

## 3. Inversion Methodology

### 3.1 Optimal Estimation

Both the MW-only retrieval and the IR + MW combined retrieval are optimal estimation-based physical inversion processes. They are used to find the geophysical state vector $\boldsymbol{X}$ for a given measurement $\boldsymbol{R}$ with

$$\boldsymbol{R} = \boldsymbol{F}(\boldsymbol{X}) + \epsilon, \tag{1}$$

where $F$ represents the radiative transfer forward model, $\epsilon$ represents the total error term that includes contributions from the measurement error, the forward model error, the representation error, and et al. A solution $\boldsymbol{X}$ is given by minimizing the cost function $J$, being defined as

$$J(\boldsymbol{X}) = \left(\boldsymbol{R} - \boldsymbol{F}(\boldsymbol{X})\right)^T \boldsymbol{S}_\epsilon^{-1}\left(\boldsymbol{R} - \boldsymbol{F}(\boldsymbol{X})\right) + (\boldsymbol{X} - \boldsymbol{X}_a)^T \boldsymbol{S}_a^{-1}(\boldsymbol{X} - \boldsymbol{X}_a). \tag{2}$$

where $\boldsymbol{S}_\epsilon$ is the covariance of the error term $\epsilon$. $\boldsymbol{X}_a$ and $\boldsymbol{S}_a$ are the background and covariance of *a priori* constraint in the state vector domain. The nonlinearity of the radiative transfer function defined in Equation (1) requires an iterative minimization process to find a solution. Following the Gauss-Newton method suggested by Rodgers [2000], a solution can be given as,

$$\boldsymbol{X}_{n+1} - \boldsymbol{X}_n = (\boldsymbol{K}^T \boldsymbol{S}_\epsilon^{-1} \boldsymbol{K} + \boldsymbol{S}_a^{-1})^{-1}\left(\boldsymbol{K}^T \boldsymbol{S}_\epsilon^{-1}(\boldsymbol{R} - \boldsymbol{R}_n) - \boldsymbol{S}_a^{-1}(\boldsymbol{X} - \boldsymbol{X}_a)\right). \tag{3}$$

Here $\boldsymbol{K}$ is the Jacobian, i.e., the first derivative which defines the sensitivity of the measurement to the input parameters,

$$\boldsymbol{K} = \left.\frac{d\boldsymbol{F}(\boldsymbol{X})}{d\boldsymbol{X}}\right|_{\boldsymbol{X}=\boldsymbol{X}_n}. \tag{4}$$

The Gauss-Newton approach works well when the degree of nonlinearity is small. The step size of the iterative process must be optimally controlled to ensure it is still within the linear region. This is achieved in SiFSAP following the method described in Wu et al. 2017 and Lynch *et al.*, 2009. The method uses the radiance residual between the observation and simulation at each step as a proxy to control the step size. Specifically, the solution is obtained by,



$$X_{n+1} - X_n = (K^T S_R^{-1} K + S_a^{-1})^{-1} \left( K^T S_R^{-1} (R - R_n) - S_a^{-1} (X - X_a) \right). \tag{5}$$

$S_R$ provides the constraint in the radiance domain that is adjusted during each step of the minimization approach. The inversion is known to be an ill-posed problem. The dimension reduction of inversion matrix is usually needed to stabilize the solution
and reduce the computational cost. The dimension reduction can be done in both the radiance $R$ and the geophysical state vector $X$ domain. While MW sounders have limited number of channels (AMSU - 15 channels; MHS - 5 channels; ATMS - 22 channels), the dimension reduction is critical to process information from hyper-spectral IR sounders' thousands of spectral channels. In NUCAPS, CHART and CLIMCAPS, only a few hundred selected IR channels are used for the retrieval (due to processing constraints in forward model and inverse model calculations). In SiFSAP, the synergetic radiance vector $R$ for the
IR+MW retrieval consists of the principal component (PC) scores of IR radiances and the channel brightness temperatures (BTs) of MW measurements:

$$R = \begin{bmatrix} Y_1 \cdots Y_{N_{ir}}, & R_1 \cdots R_{N_{mw}} \end{bmatrix} \tag{6}$$

where $Y$ denotes the PC scores of IR radiances with $N_{ir}$ being the total number of EOFs used and $R$ denotes the MW BTs of $N_{mw}$ channels. In order to reduce the dimension of the state vector $X$, atmospheric vertical profiles that are usually defined as
level (or layer) quantities on fixed pressure grid are represented as a linear combination of pre-defined trapezoidal functions in NUCAPS, CHART and CLIMCAPS. The principal component (PC) analysis is used to reduce the dimension of the geophysical variables in SiFSAP. Atmospheric profiles and surface emissivity spectra are projected onto a set of pre-computed EOFs. Table 2 lists the dimension of radiance and geophysical state vectors used in SiFSAP.

Averaging kernels for retrieved atmospheric profiles are provided in SiFSAP. The vertical resolution of the retrieved
atmospheric temperature, moisture and other trace gases can be characterized using the averaging kernel,

$$A = \left( K^T S_R^{-1} K + S_a^{-1} \right)^{-1} K^T S_R^{-1} K . \tag{7}$$

Averaging kernels are also used to derive the Degrees of Freedom (DOF) of the signal,

$$d_s = \mathrm{tr}\,(A), \tag{8}$$

a scaler used to evaluate the vertical information content provided by the measurements. Error estimations for each retrieved
variable are also included in SiFSAP output. Following the definition by Rodgers (1990), we calculate the total retrieval error covariance matrices:

$$S_x = (K^T S_R^{-1} K + S_a^{-1})^{-1} \tag{9}$$

All geophysical variables are simultaneously and directly retrieved in the SiFSAP scheme. This avoids the complicated characterization of error propagation needed in sequential retrieval algorithms, e.g., CLIMCAPS (Smith and Barnet, 2019).
The direct retrieval of state vector related to cloud properties also avoids the uncertainty introduced by 'cloud clearing', which is difficult to quantify and susceptible to the quality of the atmospheric state used to derive a clear-sky TOA spectrum for cloud clearing.

The overall QC flags are determined based on the cost function zeta ($\zeta$) that characterizes how well the simulated radiance using forward model fits the observed radiances. It is calculated as:



$$\zeta = (R^{simu} - R^{obs})^T S_\epsilon^{-1} (R^{simu} - R^{obs}) \qquad (10)$$

The zeta threshold values for MW only and IR+MW retrievals are empirically assigned to achieve an optimized balance between retrieval accuracy and yield rate. The $\zeta$ for the retrieval of trace gas species are calculated using the selected IR sounder channels in the corresponding absorption regions.

**3.2 A priori Constraint and Representation of Geophysical Variables**

*A priori* is best used in the retrieval to supplement for the information that cannot be provided by the measurements. Depending on the information content that can be obtained from IR or MW sounder data, different *a priori* constraints are used for different retrieval variables. Climatological backgrounds and error covariances used for temperature and water vapor retrieval in the SiFSAP system are derived from a combined dataset with more than thirty thousand globally-distributed atmospheric profiles (Liu et al, 2009 and Wan et al 2017). These profiles include data from European Centre for Medium-Range Weather Forecasting (ECMWF) reanalysis, radio sonde measurements, and satellite-based observations. The atmospheric profiles are represented by level quantities on a 98 pressure level grid from the surface to TOA in the retrieval system. The EOFs corresponding to the temperature and water vapor state vectors are derived from the background error covariance matrices. The surface level index, which is determined by the surface pressure value, can be quite different for different land regions while remaining relatively constant over ocean. Therefore, EOFs and *a priori* of temperature and water vapor are constructed as over-land and over-ocean groups. A conventional EOF transformation is used to represent temperature profiles in the form,

$$X_i^{Temp} = \sum_{j=1}^{98} U_{i,j}^{Temp} \cdot (P_j^{Temp} - \overline{P^{Temp}}), \qquad (11)$$

where $X_i$ represents the *i*th EOF coefficient of a corresponding temperature profile *P,* which has an unit of Kelvin, with the climatological background $\bar{P}$ being given as the mean value of the profiles included in generating the covariance matrix, and $U_i$ is the ith significant eigenvector. The water vapor EOFs are built as the logarithm of water vapor profiles,

$$X_i^{H_2O} = \sum_{j=1}^{98} U_{i,j}^{H_2O} \cdot (\log(P_j^{H_2O}) - \overline{\log(P^{H_2O})}). \qquad (12)$$

EOFs of ozone profiles are also constructed using globally distributed data but separated as over-land and over-ocean groups, similar to temperature and water vapor. The absolute value of ozone concentration in the tropospheric region is very small compared with that in the stratospheric region. In order to better represent the variational feature of ozone profiles in the tropospheric region, the ozone EOFs are built as functions of the square root of ozone profiles:

$$X_i^{O_3} = \sum_{j=1}^{98} U_{i,j}^{O_3} \cdot \left( \sqrt{P_j^{O_3}} - \overline{\sqrt{P^{O_3}}} \right). \qquad (13)$$

Moreover, *a priori* for ozone retrieval is stratified according to latitude and tropopause height to better constrain the retrieval in the regions where the ozone signal is weak. The latitude-referenced ozone climatology has been adopted as in CHART (Susskind et al. 2017) and NUCAPS (Barnet. et al. 2021), while the tropopause-referenced ozone climatology is also used for ozone retrieval studies using AIRS measurements (Wei et al., 2010) and that planned for the measurements by Tropospheric Emissions: Monitoring of Pollution (TEMPO) satellite (Johnson et al., 2018, Yang et al., 2019). The combined latitude and



tropopause information provides a quality estimate of the ozone variability that changes latitudinally and correlates with the synoptic-scale meteorological features of the tropopause. The tropopause height information is obtained from real-time forecast data provided by National Centers for Environmental Prediction (NCEP). It can also be obtained from MW-only retrievals. The readily available ozone climatology information is therefore well suited for near-real-time applications. *A priori*

for Ozone is generated using a synergistic dataset that combines data from the Model for Ozone And Related chemical Tracers (MOZART), ozone sonde measurements, the European Centre for Medium-Range Weather Forecasts (ECMWF) analysis, and the Modern-Era Retrospective analysis for Research and Applications (MERRA). The synergistic ozone profiles are binned into 18 latitudinal zones and 13 tropopause-dependent groups. To further cover the seasonal variation characteristics of the ozone climatology, the correlation relationship between the ozone profiles and the collocated temperature profiles are derived

and used for each latitude-tropopause group. The first-guess values used for the ozone retrieval can therefore be obtained using this statistically derived relationship and the temperature profiles from the MW retrieval results of the SiFSAP system. The uncertainty in those first-guess values is characterized and used as covariance constraints of the ozone *a priori* in the retrieval. Carbon monoxide (CO) EOFs are also built on the logarithm of profiles. Carbon dioxide is retrieved as averaged column density values. Methane ($CH_4$) and Nitrous Oxide ($N_2O$) are similar in a way that their concentrations are relatively stable

below the tropopause and decrease with height via various chemical processes in the stratosphere. Since $CH_4$ and $N_2O$ are well mixed in the troposphere and their mixing ratios decrease dramatically above tropopause due to chemical reactions and photolysis, their ratio profiles ($P$) can be represented as a sigmoid-like function of altitude $h$ to a good approximation,

$$P(h) = \frac{P_0}{1+e^{-(\frac{h-h_0}{a_0})}} \, , \qquad (14)$$

where $P_0$ defines the near surface mixing ratio, $h_0$ defines the dependence of vertical profiles on tropopause height, and $a_0$

determines the rate of decrement in the stratosphere. In this way, the retrieval of $CH_4$ and $N_2O$ profiles is constrained to a solution defined by three parameters. The atmospheric distributions of $CH_4$ and $N_2O$ are rather uniform zonally but exhibit a gradient with latitude. $P_0$, $h_0$, and $a_0$ values for given individual profiles are obtained by fitting the vertical profiles according to the function defined by equation (18). The first-guess values and the corresponding covariance constraints for $P_0$, $h_0$, and $a_0$ are statistically obtained using a MOZART database that includes globally distributed $CH_4$ and $N_2O$ vertical profiles of 12

different months. They are further stratified into 18×13 latitude-tropopause dependent groups. This is similar to the strategy adopted to construct the ozone *a priori* except that there is a lack of correlation between $CH_4$ or $N_2O$ profiles and the collocated temperature profiles so that the mean values of specified groups are used as the first guess values instead. The first guess of surface mixing ratio $P_0$ for each individual retrieval is further adjusted according to the globally averaged, monthly mean atmospheric methane and nitrous oxide concentration determined from the observation network of various air sampling sites

whose locations range in latitude from 90-degrees-S to 82-degrees-N (Dlugokencky et al., 1994).

The EOFs for MW surface emissivity over ocean are built from simulated emissivity spectra using the Wilheit (1979) model and an improved fast microwave water emissivity model FASTEM (Liu et al., 2011). The Masuda model (Masuda et al. 1988) and surface-leaving radiance model (Nalli et al. 2008a, 2008b) are used for the simulation of IR surface emissivity samples




over ocean. The simulations use randomly generated wind speed and surface temperature data within a realistic dynamic range.

The EOFs for MW land emissivity spectra are obtained using English's semi-empirical model (Hewison and English, 1999). The EOFs for IR land surface emissivity are constructed using data from the ECOsystem Spaceborne Thermal Radiometer Experiment on Space Station (ECOSTRESS) spectral emissivity databases (Meerdink, et al. 2019, Baldridge et al. 2009). For both MW and IR retrievals, the surface emissivity function

$$F(\varepsilon) = \log\left(\log\left(\frac{\varepsilon_{max} - \varepsilon}{\varepsilon_{max} - \varepsilon_{min}}\right)\right) \qquad (15)$$

is introduced to constrain the retrieved surface emissivity within a range between $\varepsilon_{max}$ and $\varepsilon_{min}$, which are empirically based on best knowledge of surface emissivity (Zhou et al. 2010).

Figures 4-6 demonstrate the representation of sample temperature, water vapor, ozone and carbon monoxide profiles using different numbers of EOFs as specified in Table 2. The temperature and water vapor profile samples used for the validation are from selected ECMWF reanalysis profiles. Ozone and carbon monoxide profiles are randomly selected from the

synthesized datasets used to build *a priori* constraints for the retrieval, including data from sonde measurements, reanalysis databases, and geochemical model results. Along with the plots that illustrate the distribution of true profiles, the EOF representation errors are quantified in terms of their mean bias and root-mean-square (RMS) values. Figure 7 demonstrates the representation of sample $N_2O$ and $CH_4$ vertical profiles from MOZART using the sigmoid representation functions.

**3.3 Bias correction**

As compared with the traditional IR+MW algorithms that rely on cloud-clearing, the SiFSAP algorithm fits the TOA radiance directly and maximizes the contribution from the measurement-provided information. The accuracy of the retrieval critically depends on how well the forward model errors are addressed. The correction for forward model errors (here referred as 'bias correction') in the SiFSAP scheme includes two parts: 1) the correction for the channel brightness temperatures of MW sounder measurements; 2) the correction for the hyperspectral measurements of IR sounders. Forward model errors, which can be

generalized as the difference between the simulated radiance and the observations, may arise from the spectroscopy inaccuracies, and/or the fast parameterizations used in the radiative transfer models. In an optimal estimation-based retrieval scheme, forward model errors can be corrected by subtracting the systematic bias (the mean value of $\epsilon$ defined in Equation 1) from the observation and accommodating the uncertainty in the error covariance of radiance residuals after the subtraction ($S_\epsilon$ defined in Equation 6).

Estimation for the systematic bias and the error covariance is done by comparing the observations with radiances computed by the forward models using the best estimate of the truth as inputs. A common practice is to use the reanalysis data which are spatiotemporally matched to the selected ensemble of satellite observations as the truth of inputs. Data from ECMWF reanalysis has been used to evaluate the simulation of MW sounders like ATMS (Zhou, Y. and Grassotti, C. 2020) and MHS (Schulte et al. 2019). Aumann et al. (2018) used ECMWF data to evaluate the simulation of hyperspectral sounder

measurements under cloudy sky conditions using various radiative transfer models (RTMs) with cloud scattering simulation



capability, including PCRTM. All RTMs fit reasonably well in the 11-μm atmospheric window area. PCRTM has the smallest bias among the 6 RTMs for the cloudy sky observations at 900 cm⁻¹ and provides best match with observed AIRS radiances in shortwave IR spectral region where the solar scattering of clouds are important.

MW sounder measurements are known to have systematic, scan-angle dependent errors due to effects of antenna side-lobes not being adequately accounted for in the calibration process. The differences between measured and computed spectra are usually scene dependent. Therefore, dynamic bias correction schemes for MW measurements have been implemented in the numerical weather prediction (NWP) data assimilation (DA) systems (Zhu et al. 2014, Dee et al. 2009) and the physical retrieval systems (Schulte et al. 2019). The dynamic bias correction schemes rely on the pre-trained relationship, being either a regression-based linear or neural network-based nonlinear scheme, between the radiance bias and the predictors. The

predictors include satellite angles, atmospheric and surface properties collocated with observations. Zhou and Grassotti (2020) studied the use of the ATMS brightness temperature (BT) as the major predictor in the Microwave Integrated Retrieval System (MiRS, https://www.star.nesdis.noaa.gov/mirs). BTs are used along with other observation angle and scene dependent predictors including latitude, cloud liquid water, total precipitable water, and surface skin temperature in the bias correction scheme. In the SiFSAP scheme, the MW bias corrections are stratified for different scan angles. Considering that the

information about atmospheric and surface properties is already embedded in MW spectra, we choose the MW spectra as the only predictor in order to facilitate the operation of the SiFSAP algorithm, especially for near-real-time data production. The bias prediction used for the MW sounder is implemented through the following equation:

$$\epsilon_j^\theta = \sum_{i=1}^{Nmw} A_{ji}^\theta R_i^\theta, \tag{16}$$

where $j$ and $i$ are MW sounder channel index numbers; $\epsilon_j^\theta$ is the scan angle dependent bias in brightness temperature of MW

sounder channel $j$; $A_{ji}^\theta$ is the regression prediction coefficient that links the bias to the MW channel measurement $R_i^\theta$; and $A_{ji}^\theta$ is trained using the least-square fit on the training sample. The matchup training samples of $\boldsymbol{\epsilon}^\theta$ and $\boldsymbol{R}^\theta$ are constructed using collocated ECMWF data and MW sounder measurements from selected 'focus' days. ECMWF does not provide surface emissivity and accurate cloud information. Therefore, emissivity is tuned along with cloud properties within the constraint defined by the preconstructed *a priori*. The solutions that provide the best match to the observations are selected. The

difference between $\boldsymbol{R}^\theta$ and the corresponding fitted radiances (in BT) is the bias $\boldsymbol{\epsilon}^\theta$. We filter out the outliers of the matchup samples where the absolute differences between the simulated MW brightness temperatures using reanalysis data and the observed ones are greater than a predetermined threshold. Figure 8 illustrates the scan-angle dependent bias of ATMS measurements onboard of SNPP and NOAA-20(JPSS-1) satellites, respectively. Figures 9 and 10 further demonstrate the probability density distribution of brightness temperature difference between the observations and the simulations for different

ATMS channels. Figures 9 and 10 also show that the scene-dependent biases can be effectively corrected using the regression-prediction scheme. It is noted here that the global mean daily biases from the simulation cannot be characterized as static offsets. The magnitudes of those offsets for different days can be very different and therefore cannot be effectively corrected via a static offset subtraction.



The bias correction for the IR hyperspectral retrieval follows a regression-prediction approach similar to that for the MW
retrieval. The biases are usually not scan angle dependent so that a unified correction is used for measurements at different
satellite scan angles:

$$\epsilon_j = \sum_{i=1}^{N_{eof}} A_{ji} R_i. \tag{17}$$

Here the bias and radiances are presented in the EOF domain. $N_{eof}$ is the number of EOFs used to represent the hyperspectral
sounder radiances. Again, we need to fit surface spectral emissivity and cloud properties to minimize the differences between
the simulated spectral radiances and the corresponding sample observations. The static bias correction term $\epsilon_j$ is small. What
is critical here is the magnitude and the distribution of spectral fitting residuals, which define the error covariance used for the
retrieval ($S_\epsilon$ in Equation 2). Figure 11 plots the spectral error covariance used for the SNPP-CrIS retrieval.

## 4. Results and Assessment

### 4.1 Radiance Fitting Assessment

One important quality assessment factor for sounder retrieval products is how well the retrieved properties fit the radiance
measurements. Providing the best fit to the measured TOA radiances is the first important indicator of the correct utilization
of maximized information provided by the measurement. The capability of providing radiance 'closure' justifies the retrieval
products' application for climate monitoring. It is especially critical to ensure the traceable accuracy when the data from
multiple sounder measurements like AIRS, CrIS and IASI are fused together to establish a long-term climate data record
(Strow et al., 2021; Wu et al., 2020).  Figure 12 shows the global mean spectral fitting residuals between the CrIS radiances
from SiFSAP and the observations for January 14, 2016. This daily mean fitting residual is derived using ~90% of CrIS single
FOV measurements of that day. The capability of fitting the single FOV measurements under cloudy sky conditions greatly
facilitates the use of SiFSAP for climate studies that requires high spatial resolution and all-sky sampling.

### 4.2 Temperature and Water Vapor Profiles

The validation of temperature and water vapor profiles from SiFSAP has been done using results from selected testing
days. Figures 13 and 14 plot the global mean and RMS values of the differences between the temperature and the water vapor
retrieved from SNPP-CrIS/ATMS measurements during July 16th of 2017 and that from the collocated ECMWF data. The
bias and RMS for the temperature difference are calculated as:

$$T_{\text{Bias}}= mean(T_{SiFSAP} - T_{ECMWF}) , \quad T_{\text{RMS}}= \sqrt{mean((T_{SiFSAP} - T_{ECMWF})^2)} . \tag{18}$$

The bias and RMS for the water vapor are calculated as:

$$H_2O_{\text{Bias}}= \frac{mean(H_2O_{SiFSAP}-H_2O_{ECMWF})}{mean(H_2O_{ECMWF})} \times 100\% , \quad \text{RMS}= \frac{\sqrt{mean((H_2O_{SiFSAP}-H_2O_{ECMWF})^2)}}{mean(H_2O_{ECMWF})} \times 100\% . \tag{19}$$

We can see that above 10 km, temperature profiles from SiFSAP have a better than 1 K retrieval accuracy. The retrieval
uncertainty becomes larger in the lower troposphere region, mostly due to the limited sensitivity of IR sounders to atmospheric



profiles below thick clouds. The retrieval accuracy for profiles below clouds becomes more dependent on the retrieval accuracy

of the MW sounders as clouds get thicker. As compared with over ocean retrievals, the relatively larger uncertainty in the land surface emissivity leads to a larger uncertainty in the near surface temperature retrieval. The relative error of water vapor retrieval is around 20% or smaller in the complete tropospheric region.

## 4.3  Surface emissivity

Figure 15 demonstrates sample land surface emissivity spectra retrieved from CrIS observations over different areas with

different surface conditions. We can clearly see the strong spectral feature in the quartz reststrahlen band between 8 µm and 10 µm (1000 cm$^{-1}$ ~ 1250 cm$^{-1}$) for samples in the desert and very different emissivity features for surfaces of soil and/or plants.   Figure 16 compares the land surface emissivity at 11 µm of two selected days (January 14, 2016 and August 8, 2017) with the Aqua MODIS daily emissivity from MOD21 Land Surface Temperature and Emissivity product (Hulley et al. 2016). The difference between subplots A2 and A1 illustrates the change of surface emissivity that reflects the seasonal change

(January - August) of vegetation coverage. There is a clear correlation between the emissivity change and the vegetation coverage change shown in subplots C1 and C2 as the normalized difference vegetation index (NDVI). The NDVI values are extracted from MODIS/Terra Vegetation Indices Monthly L3 Global 0.05 Degree Climate Modeling Grid product (Didan and Huete, 2015). There is a noticeable emissivity difference between subplots A1 and A2, which can partly be explained by the change of snow coverage in this area from January to August in 2016 (shown in subplots D1 and D2). The snow coverage data

are extracted from daily Level-3 (L3) MODIS/Aqua snow coverage data products that provides the percentage of snow-covered land observed daily within 0.05° (approx. 5 km) MODIS Climate Modeling Grid (CMG) cells (Hall and Riggs, 2021).

## 4.4  Trace gases: O3, CO, CO2, CH4, N2O

The SiFSAP atmospheric composition products include the retrieved volume mixing ratio of $CO_2$ $O_3$, CO, $CH_4$, $N_2O$ at 98 vertical pressure level grids defined by the PCRTM algorithm. SiFSAP products include trace gas profiles for each sounder

FOV, i.e., matching the native spatial resolution of hyperspectral sounder instruments, SiFSAP $O_3$ data have been used to study stratospheric intrusion (Xiong et al., 2022A) and cold air outbreaks (Xiong et al., 2022A). SiFSAP CO data have been used for process-oriented analysis of emission from large wildfires and air pollution transport studies (Xiong et al., 2022C). The validation and further developments of those atmospheric composition products have been an on-going effort. Sample validation studies are presented here to illustrate the overall performance of SiFSAP.

Figure 17 demonstrates the inter-comparison study of satellite-based CO observation on/around May 12$^{th}$, 2020 between SiFSAP of SNPP/CrIS, Metop-B/IASI daily CO product, and CO data from the Measurement of Pollution in the Troposphere (MOPITT) on board the Terra satellite. IASI CO data are generated using the Fast Optimal Retrievals on Layers for IASI (FORLI) software (Hurtmans et al. 2012). IASI measures TOA spectral radiances between 645 and 2760 cm$^{-1}$ with a 0.25 cm$^{-1}$ spectral interval between adjacent channels. CO vertical profiles are retrieved using the spectral channel

measurements between 2128 and 2206 cm$^{-1}$. As a comparison, SNPP/CrIS lacks the spectral coverage between 2128 and 2155



cm$^{-1}$ and only provides spectral measurement with a 0.625 cm$^{-1}$ spectral interval. However, the ultra-low instrument noise of CrIS in the CO absorption region improves the information content that allows the capture of key features of the source and sink climatology of CO. The spatial and vertical distribution of CO concentration from SiFSAP in the middle-to-upper troposphere region agree well with FORLI CO data, which also generally agree with the MOPITT CO data

(NASA/LARC/SD/ASDC, 2000). MOPITT measures CO using the near-infrared (NIR) band near 2.3 μm and the thermal-infrared (TIR) band near 4.7 μm. As compared with the swath width of CrIS and IASI that is around 2200 km, the swath width of MOPITT observations is only around 640 km, which can only allow a global coverage of CO measurements on a weekly basis. Therefore, the MOPITT CO data of multiple days (from May 10th to May14th, 2020) are plotted together to have a better global scale visualization. The total column amounts of SiFSAP CO from SNPP-CrIS agree better with FORLI data in

terms of spatial distribution at global scale. IASI FORLI results give much larger total column amount than that from both SNPP-CrIS SiFSAP and MOPITT. SNPP-CrIS SiFSAP CO data agree better with MOPITT data in terms of the scale of the total column amount over high CO concentration areas, but there is an obvious difference in spatial distributions which cannot be simply ascribed to the temporal difference between two observations. IR sensors are known to have limited sensitivity close to the surface due to the generally low thermal contrast between the ground and the air above it. MOPITT CO product is

supplemented with enhanced surface CO mixing ratio from *a priori* based on the Community Atmosphere Model with Chemistry (CAM-chem, Buchholz et al. 2019). Consequently, the spatial distribution of total column amount of MOPITT CO is strongly correlated with the surface CO distributions, which is not the case in SNPP-CrIS SiFSAP and FORLI CO products. The validation of the CAM-chem based *a priori* and its impact on the CO retrieval in the lower troposphere to surface region needs to be further studied.

Figure 18 compares the total column of O$_3$ on September 19$^{th}$, 2019 from SNPP-CrIS SiFSAP with that from SNPP-CrIS CLIMCAPS, SNPP-OMPS (Jaross, G., 2017), and MetopB FORLI daily O$_3$ product. The ozone hole over the Antarctica region is clearly captured by all products. It is noted here that IR sensors like CrIS, AIRS, and IASI are generally more sensitive to the ozone distribution in the upper troposphere while ultraviolet measurements like OMPS are more sensitive to stratospheric ozone. Both instruments can measure the tropospheric columns but lack vertical sensitivity in the troposphere (Fu et al. 2018).

The results from two products (SiFSAP and CLIMCAPS) using the same sounder measurements agree well over most of the area.

Figures 17 and 18 show that the SiFSAP system works effectively under all sky conditions. As compared with FORLI that only provides CO and O$_3$ data for cloud free or almost clear (with a less than 13% cloud fraction) observations (George et al., 2009, Boynard et al., 2018), the capability of accounting for the cloud scattering in the SiFSAP algorithm ensures a much

higher retrieval yield rate. Although CLIMCAPS can retrieve CO for most of the observations under cloudy sky conditions, it fails in area under overcast skies (shown as white area in Figures 17 and 18) because the lack of contrast between observations of adjacent FOVs impose challenges on the implementation of the cloud clearing method.

Validation to CO$_2$, N$_2$O, and CH$_4$ from SiFSAP is very limited and remains to be completed in the near future. Therefore, these products are still subject to more research and improvement.





## 4.5 Cloud properties


Cloud optical depth, particle size, and cloud height (represented by the cloud top temperature CTT) are simultaneously retrieved along with other geophysical variables in the SiFSAP algorithm. Details about the cloud scattering model can be found in previously published PCRTM and physical retrieval algorithm papers (Liu et al. 2006, Liu et al. 2007, Liu et al. 2009, Wu et al. 2017). Cloud properties for one individual CrIS footprint are retrieved under the assumption of one effective single

layer with the cloud transmittance, reflectance, and emissivity defined by the optical depth and the particle size. Ice and water clouds are discerned based on the overall spectral characteristics of the cloud emissivity (transmittance). In the iterative retrieval process, both cloud phase options are tried and the one providing the best spectral fitting is saved as the solution. Earlier simulation studies have shown that the cloud phase can be retrieved with a very high accuracy rate (>95%) if the hyperspectral feature of the ice and water clouds can be fully explored (Wu et al. 2017).

Cloud properties from hyper-spectral sounder measurements can be validated using the collocated imager observations like MODIS or VIIRS (e.g. Yue et al. 2022). The VIIRS Atmosphere L2 Cloud Properties Product (Platnick et al., 2017, Heidinger and Li, 2017) is used to validate the cloud properties from SiFSAP of SNPP CrIS. Since VIIRS does not have IR channels in the 13 μm $CO_2$ absorption band, the MODIS $CO_2$ slicing solution for cloud top pressure retrievals for cold clouds is replaced with an IR window channel optimal estimation approach coupled with a Cloud-Aerosol Lidar and Infrared Pathfinder Satellite

Observations (CALIPSO)-derived *a priori* (Heidinger et al. 2019). The CTT of SiFSAP CrIS is compared with the average values of the CTT of VIIRS pixels within the CrIS footprints as shown in Figure 19. The global scale spatial distribution of CTT from SiFSAP agrees well with that from the VIIRS cloud product except in the Arctic region where CTT retrieved from CrIS measurements tends to be warmer than VIIRs results. The correlation coefficient between the VIIRS CTT and CrIS CTT shown in Figure 19 is larger than 0.93. Uncertainty in retrieved cloud properties tends to be larger for very thin clouds due to

the challenge of extracting weak IR spectral signatures embedded in the measurement or forward simulation errors. Figure 19 only shows results with retrieved cloud optical thickness larger than 0.4 (cloud emissivity larger than 0.1) to better illustrate the retrieval accuracy when there is adequate measurement-provided information. A larger than 0.8 correlation coefficient can still be achieved even when we include more thin cloud footprints with optical depth as small as 0.05.

Direct comparison between the effective cloud optical depth (COD) and the effective particle radius (Re) retrieved for an

individual CrIS FOV and the corresponding mean values for the collocated VIIRS pixels within the CrIS FOV can be challenging due to several factors. First, the spatial heterogeneity among VIIRS pixels means the IR radiative contribution from a cloud layer with an averaged VIIRS COD can be very different from the combined contribution from individual VIIRS pixels due to the nonlinear nature of the radiative transfer

$$F(\overline{\text{COD}}, \overline{\text{Re}}) \neq \sum_{i=1}^{N} \frac{F(\text{COD}_i, \text{Re}_i)}{N}, \qquad (20)$$

where $F$ is the forward operator. Second, the inconsistency between the cloud scattering models used for sounder retrieval and for imager retrieval can further introduce large biases or uncertainties between two sets of COD and Re. Third, inconsistency can also arise from a lack of consistency and accuracy in the atmospheric and surface state assumed for the cloud property



retrievals. As compared with COD and Re, it is relatively more straightforward to compare the effective cloud emissivity (fraction) values retrieved from CrIS and VIIRS measurements. The CrIS FOV cloud emissivity can be related with the VIIRS

pixel effective cloud emissivity and the corresponding spatial fraction under the assumption that the total thermal emissions measured by CrIS and VIIRS within the same spectral band are consistent:

$$B_\nu(T_C^{CrIS})\epsilon_C^{CrIS} + B_\nu(T_S)\epsilon_S(1 - \epsilon_C^{CrIS}) = f\sum_{i=1}^{N}\epsilon_{C,i}^{VIIRS}B_\nu(T_{C,i}^{VIIRS}) + (1 - f\sum_{i=1}^{N}\epsilon_{C,i}^{VIIRS})B_\nu(T_S)\epsilon_S, \quad (21)$$

where $B_\nu$ represents the Planck function at wavenumber $\nu$, $\epsilon_C^{CrIS}$ and $T_C^{CrIS}$ are CrIS cloud emissivity and CTT, $\epsilon_{C,i}^{VIIRS}$ and $T_{C,i}^{VIIRS}$ are cloud emissivity and CTT of individual VIIRS pixels, $f$ represents the spatial fraction of VIIRS cloud pixels within

a CrIS FOV, $T_S$ and $\epsilon_S$ are surface skin temperature and surface emissivity which are assumed to be homogeneous within a single CrIS FOV. Equation 21 is justified under the condition that $\nu$ is within a 'window' spectral region where atmospheric absorption and thermal emission can be neglected, and the effective cloud reflectivity is close to zero. Therefore, the cloud transmissivity is approximated as $1 - \epsilon_C$. A more simplified form can be used

$$\epsilon_C^{CrIS}T_C^{CrIS} + (1 - \epsilon_C^{CrIS})T_S = \sum_{i=1}^{N}f_i\epsilon_{C,i}^{VIIRS}T_{C,i}^{VIIRS} + (1 - f\sum_{i=1}^{N}\epsilon_{C,i}^{VIIRS})T_S \quad (22)$$

by utilizing the fact that the Planck function is linear enough and the surface emissivity is close to unity. Such an approach to check the radiometric consistency between cloud properties from IR sounders and imagers has been used in the AIRS-MODIS cloud retrieval validation study (Kahn et al. 2007, Nasiri et al. 2011). Figure 20 demonstrates the comparison between the effective brightness temperature of CrIS $T_{eff}^{CrIS}$ and that of corresponding VIIRS measurements $T_{eff}^{VIIRS}$, with the definition being given as

$$T_{eff}^{CrIS} = \epsilon_C^{CrIS}T_C^{CrIS} + (1 - \epsilon_C^{CrIS})T_S, \quad (23)$$

$$T_{eff}^{VIIRS} = f\sum_{i=1}^{N}\epsilon_{C,i}^{VIIRS}T_{C,i}^{VIIRS} + (1 - f\sum_{i=1}^{N}\epsilon_{C,i}^{VIIRS})T_S. \quad (24)$$

A good agreement is found between $T_{eff}^{CrIS}$ and $T_{eff}^{VIIRS}$ except a small percentage of samples in the Arctic. The cloud emissivity data used for this study are the retrieval results based on the NOAA Daytime Cloud Optical and Microphysical Properties (DCOMP; Walther and Heidinger, 2012) algorithm. Li et al. (2020) found that VIIRS cloud data products tend to have larger

uncertainties in polar regions due to the lack of VIIRS spectral measurements in IR water and $CO_2$ absorption channels. They found a major improvement for the cloud mask can be achieved over polar regions by fusing the collocated CrIS measurements in the missing spectral region with the VIIRS data. Although the radiative consistency between cloud properties from SiFSAP CrIS and that from VIIRS is high, the surface skin temperature $T_S$, CTT, and the cloud effective emissivity (fraction) are highly correlated with each other. Uncertainties in either $T_S$ or CTT will introduce inconsistency between the effective emissivity

from these two measurements. Even though the three parameters compensate each other to fit radiometrically to the observations, the effective cloud emissivity from CrIS and VIIRS measurements can still be quite different. This is especially the case when there is a lack of thermal contract between $T_S$ and CTT.



## 4.6 Averaging kernels

Hyperspectral sounder measurements provide rich information for temperature and humidity vertical profiling. Figures 21 and
22 demonstrate typical temperature and water vapor retrieval averaging kernels from SNPP CrIS SiFSAP. Figure 21 clearly
shows that high vertical resolution temperature retrieval can be achieved by the SiFSAP algorithm even in the lower
tropospheric region near the surface. The sum of the averaging kernel rows, also known as 'verticality', is usually used to
characterize how much information comes from the measurements. A verticality value close to one means measurement
provides dominant information so that a retrieval system's dependence on the *a priori* is minimized. On the other hand, a
verticality value close to zero indicates that the system is heavily dependent on the *a priori* since the measurement does not
provide much information. Figure 21 shows that hyperspectral measurements can well resolve the temperature profile from
troposphere to stratosphere under a clear sky condition. The information from the measurements degrades in the lower
troposphere under a cloudy condition region due to the weaking of thermal emission signal by clouds. The averaging kernel
for water vapor retrieval (Figure 22) is relatively less sensitive to clouds, but the information provided by hyperspectral
measurements to retrieve water vapor in upper troposphere and stratosphere region is limited.

As compared with temperature and water vapor, the measurement information from hyperspectral IR sounders for trace gases
retrieval is relatively limited and more scene dependent. Figure 23 shows the averaging kernels of $O_3$ retrievals from SNPP
CrIS SiFSAP of September 20[th], 2019 for different latitudinal regions.  The $O_3$ retrieval has sensitivity peaks in both
stratosphere and upper troposphere. The measurement sensitivity for lower tropospheric $O_3$ is the highest in the tropical region
and tends to decrease as the observations move to higher latitude regions. Overall, the SiFSAP system provides decent vertical
resolution of $O_3$ profiling based on real CrIS observation data, which is comparable to what has been demonstrated for IASI
measurements via an end-to-end simulation study (Wu et al. 2017).

Samples averaging kernels from SiFSAP CO product are shown in Figure 24 for different latitudinal bands, as CO retrieval is
more latitudinal dependent as compared with $O_3$. CrIS full spectral resolution measurements provide abundant information for
the tropospheric CO retrieval in the tropical region (with verticality close to 1). The measurement information becomes less
dominant in the mid-latitude region and very limited in the polar regions. This is partly due to the fact that thermal emission
signals due to CO absorption in the atmosphere are weaker in lower temperature region. Ultimately, the total measurement
sensitivity of CO is limited by the CrIS instrument noise level in the CO absorption spectral region. The vertical resolution of
CO retrieval is very limited in the current version of SiFSAP. Similar to SiFSAP, the reported vertical resolution of CO retrieval
in other IR sounder retrieval systems, e.g. FORLI, AIRS CO retrieval, and CLIMCAPS (George et al. 2009, Smith and Barnet,
2020), is also very limited. This can be ascribed to the weak thermal contrast among signals in CO measurement channels of
IR sounders and the vertical distribution constraints from the *a priori* that remain to be optimized.





## 5. Conclusions and Future Work

SiFSAP retrieval algorithm has been developed to generate a high spatial resolution and radiometrically consistent hyperspectral sounder product to explore the applications of sounder observations in areas that have not been fully addressed by the current operational sounder products. SiFSAP products include temperature, water vapor, $O_3$, $CO_2$, CO, $CH_4$, and $N_2O$ profiles, as well as surface properties (including surface skin temperature and surface emissivity) and cloud properties (including cloud top pressure, height, temperature, effective cloud optical depth, and effective cloud particle size). Following an optimal estimation scheme and the efficient and accurate forward radiative transfer model PCRTM, SiFSAP also provides users with the averaging kernels and error estimates to facilitate better uncertainty quantification in physical process studies and data assimilations using sounder products, as well as intercomparisons of multiple observational and model products.

Initial validation on key SiFSAP Level 2 variables has been carried out using SNPP-CrIS as an example. More extensive studies and validation of SiFSAP products for other satellites will be conducted. Validation for $CO_2$, $CH_4$, and $N_2O$ has been initiated but a lot of work remains to be done, so these three trace gas products are released as exploratory data products at the current stage.

One key advantage of the SiFSAP algorithm is its applicability for multiple IR and MW sounder systems. In addition to CrIS/ATMS onboard SNPP and JPSS satellites, the SiFSAP system is ready for the processing of both AIRS/AMSU and IASI/AMSU/MHS data. Simulations based end-to-end studies and some evaluation work using sample IASI data have been demonstrated (Wu et al. 2017, Liu et al. 2009). Some AIRS retrieval case studies using the SiFSAP algorithm have already demonstrated the advantage of SiFSAP over traditional AIRS Level 2 product in capturing the high spatial resolution feature of gravity wave signals in the stratospheric temperature (Wu et al. 2019). SiFSAP provides a solution to retrieve key climate variables from different hyperspectral sounder observations using a consistent physical algorithm. This is not only important for effectively fusing information from multiple instruments, but also essential to constructing a long-term continuous climate data record from the Program of Record sounder observations. The capability of using a unified radiative transfer model (i.e., PCRTM) to accurately fit the spectral radiances measured by all modern era operational hyperspectral sounders under all sky conditions is essential for the climate trend/anomaly retrieval study from a radiometric consistency perspective.

SiFSAP will support weather and atmospheric dynamics studies by providing high spatial resolution temperature and water vapor profiles that can be used to reveal mesoscale atmospheric variations. The algorithm's capability of using the spectral information from all hyper-spectral channels via PC analysis makes it easy to be adapted and affordable for future sounder applications with a much higher spectral resolution and much more channels (e.g. IASI-NG). The scheme requires minimal auxiliary data to provide the *a priori* constraints and is suitable for real-time and environmental monitoring applications. Future work includes exploring the SiFSAP algorithm's application potential in challenging areas (e.g. Planetary Boundary Layer studies) by further improving the utilization of spectral information and the accommodation for forward model errors. The development of a long-term climate record based on SiFSAP using the climate spectral fingerprinting scheme is also underway.



**Data availability.** SiFSAP will soon be available to public from the NASA Goddard Earth Sciences Data and Information Services Center (GES DISC). The availability of the SNPP SiFSAP data is currently on a request basis. The SNPP CLIMCAPS data is available from GES GISC (https://10.5067/62SPJFQW5Q9B).VIIRS cloud property data is available from Level-1 and Atmosphere Archive & Distribution System Distributed Active Archive Center (LAADS DAAC  https://ladsweb.modaps.

eosdis.nasa.gov/search/order/1/CLDPROP_L2_VIIRS_SNPP—5111). AQUA MODIS monthly land surface emissivity data is available from the Land Processes Distributed Active Archive Center (LP DAAC https://lpdaac.usgs.gov/ products/myd11c3v006/). AQUA MODIS monthly vegetation index data is available from LP DAAC (https://lpdaac.usgs.gov/ products/myd13c2v006/). Snow coverage data is from NASA National Snow and Ice Data Center Distributed Active Archive Center (https://doi.org/10.5067/ MODIS/MYD10C1.061). METOP-B IASI O3 and CO data are available from IASI portal

(https://iasi.aeris-data.fr/o3/, https://iasi.aeris-data.fr/co/). MOPITT CO data is from the NASA Atmospheric Science Data Center (ASDC) (ftp://l5ftl01.larc.nasa.gov/MOPITT/).  OMPS O3 data is from NASA Earth Data (https://omisips1.omisips. eosdis.nasa.gov/outgoing/OMPS/LANCE/NMTO3-L2-NRT/).

**Author contributions.** WW and XL wrote the manuscript with discussion and editing inputs from XX DZ QY[3] AL QY[2] and

560 LL; The SiFSAP algorithm and PCRTM were developed by XL and WW; XL QY[2] and WW are responsible for the update of PCRTM.WW and LL produced CrIS/ATMS SiFSAP results; WW collected CLIMCAPS, IASI, MOPITT, MODIS and OMPS data used for the validation study. QY[3] provided VIIRS cloud property data collocated with CrIS observations. WW carried out the validation study with suggestions and technical support from XL and XX and LL.

**Competing interests.** The contact author has declared that none of the authors has any competing interests.

**Acknowledgement.**  We are grateful for the data production and storage technical support provided by the team members responsible for the Science Investigator-led Processing System (SIPS) at Jet Propulsion Lab, California Institute of Technology. We thank NASA Advanced Supercomputing (NAS) facility for providing the computational resources and the

570 corresponding technical support. We also thank CRTM technical support team for providing the software and the instruction. This work was funded by the NASA 2017 Research Opportunities in Space and Earth Sciences (ROSES) solicitation NNH17ZDA001N-TASNPP and the NASA 2020 ROSES solicitation NNH20ZDA001N.

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

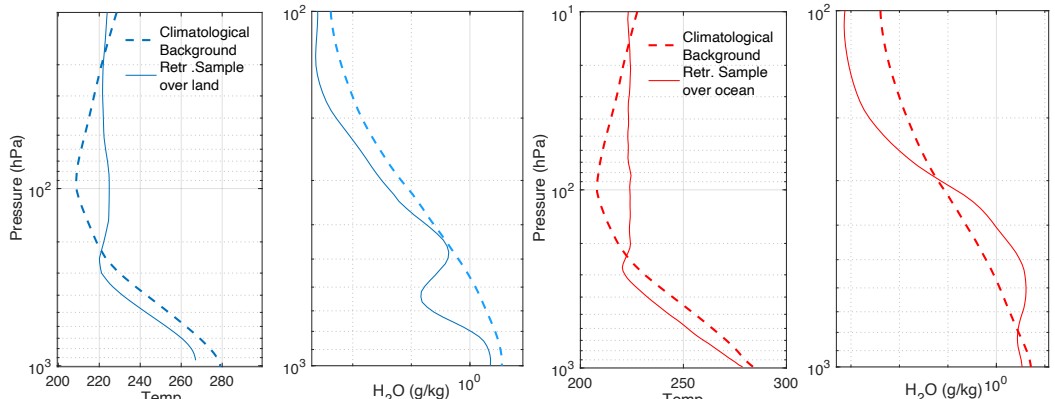

**Figure 1 Climatological background used for temperature and water vapor retrievals (dash curves) in the SiFSAP algorithm. The final retrieval results (sample retrieved profiles presented as solid curves) can be very different from the background values.**


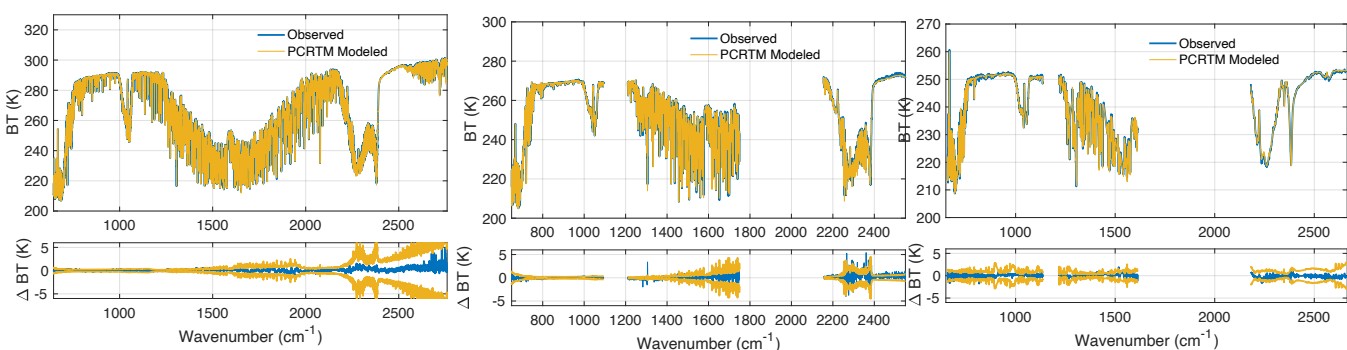

**Figure 2 IR sounder radiances fitted by SiFSAP. Left: IASI; Middle: CrIS; Right: AIRS. The radiance fitting residues (blue curves in lower subplots) are compared with the instrumental random noise (with the magnitude being marked using yellow curves).**



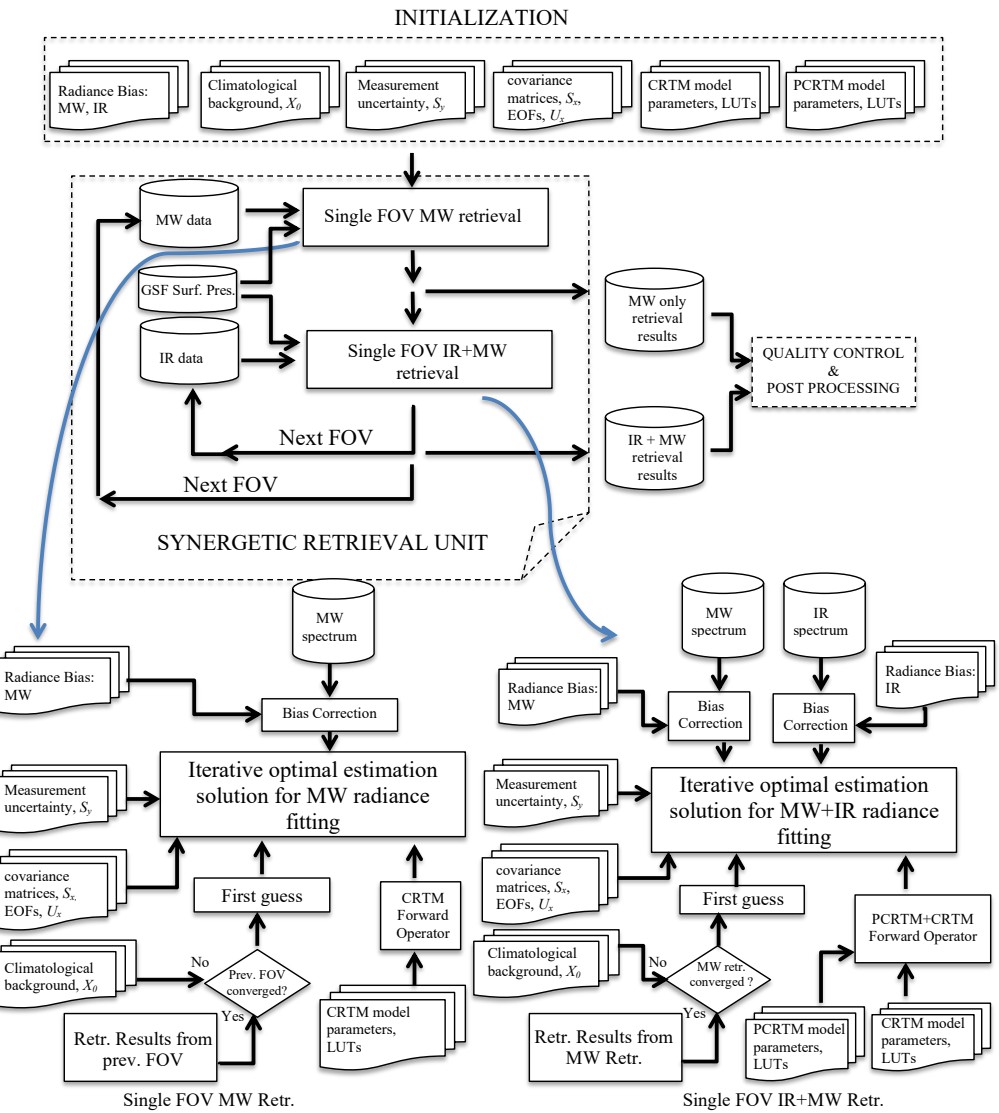

**Figure 3 Flow diagram of the SiFSAP data processing scheme.**






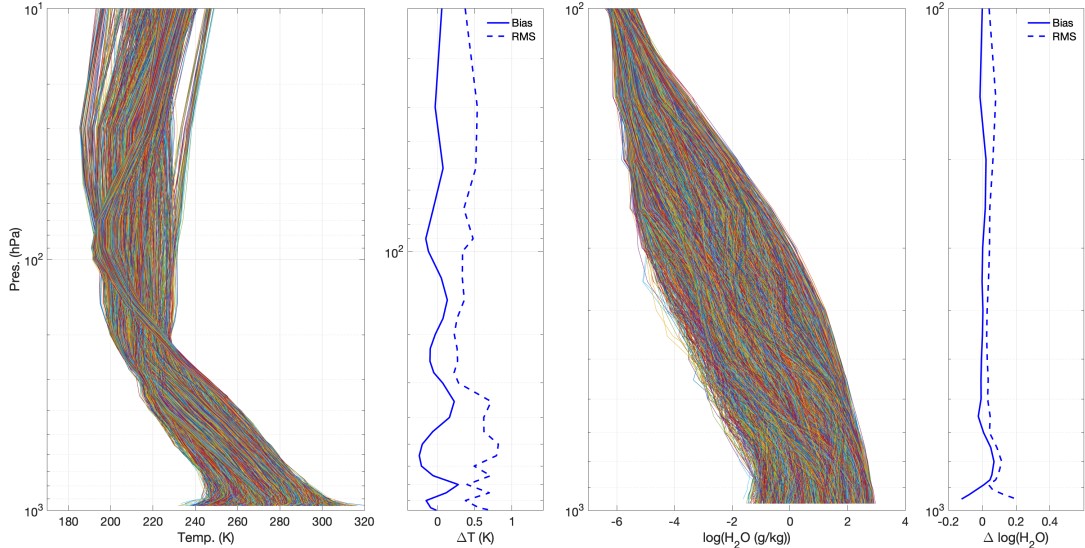

**Figure 4 EOF representation errors of temperature and water vapor profiles. (a) and (c) the original temperature and water vapor profiles; (b) and (d) the mean and the RMS values for the difference between the original profiles and the profiles being represented using limited number of EOFs (20 EOFs are used for temperature profiles and 15 EOFs are used for water vapor).**

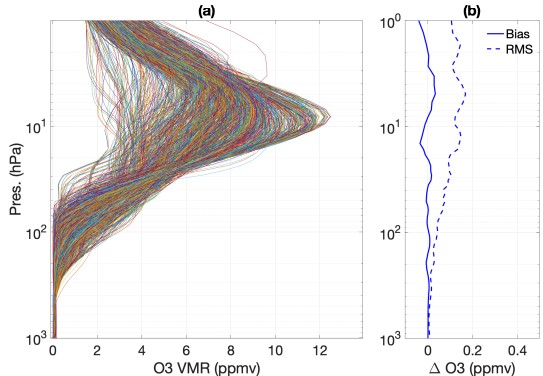

**Figure 5 EOF representation errors of ozone profiles. (a) the original ozone profiles; (b) the mean and the RMS values for the difference between the original profiles and the profiles being represented using 10 EOFs.**




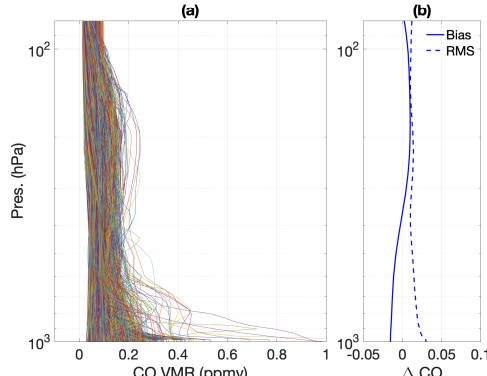

**Figure 6 EOF representation errors of carbon monoxide profiles. (a) the original ozone profiles; (b) the mean and the RMS values for the difference between the original profiles and the profiles being represented using 4 EOFs.**

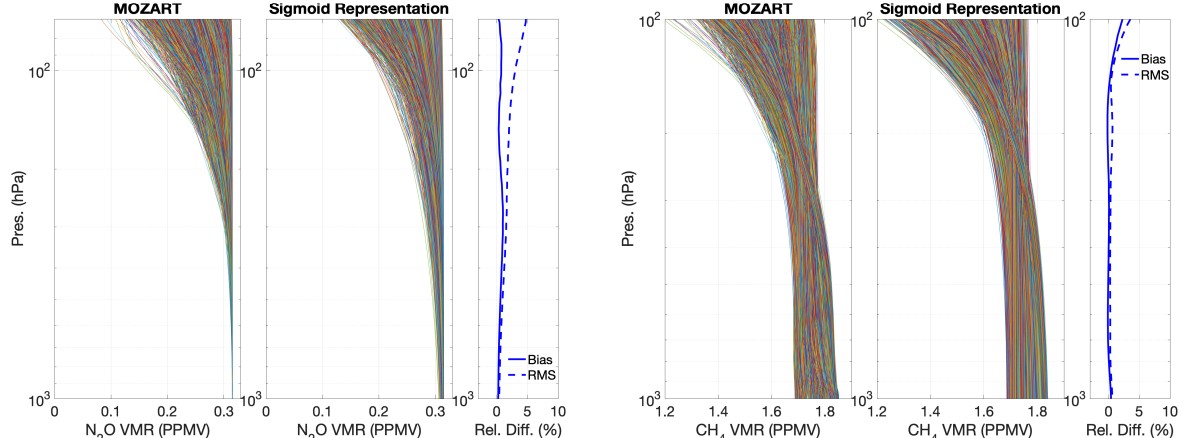

**Figure 7 The representation of nitrous oxide and methane profiles using sigmoid functions.**





**Table 1.** Geophysical parameters included in SiFSAP.

|  | From IR+MW Synergistic retrieval | From First Step MW only retrieval |
|---|---|---|
| [1]Temperature Profile (K) | Yes | Yes |
| [1]Water Vapor MMR profile (g/kg) | Yes | Yes |
| [1]$CO_2$ VMR profile (ppmv) | Yes | |
| [1]$O_3$ VMR profile (ppmv) | Yes | |
| [1]$CH_4$ VMR profile (ppmv) | Yes | |
| [1]CO VMR profile (ppmv) | Yes | |
| [1]$N_2O$ VMR profile (ppmv) | Yes | |
| Surface Skin Temperature (K) | Yes | Yes |
| [2]IR Surface Emissivity | Yes | |
| [3]MW Surface Emissivity | Yes | Yes |
| Effective Cloud Top Pressure (hPa) | Yes | |
| Cloud Particle Size (μm) | Yes | |
| Cloud Optical Thickness | Yes | |
| Cloud phase (ice or water) | Yes | |
| Liquid Water Content (g/m$^2$) | Yes | Yes |

[1] Atmospheric profiles are given at 98 pressure levels;

[2] IR surface emissivity at native mono-frequency bins defined by the PCRTM are provided. Number of frequency bins for different sounders are: AIRS - 500; IASI- 753; CrIS at full resolution - 540; CrIS at nominal resolution – 485;

[3] MW surface emissivity are given for each channel of MW sounders.







**Table 2.** Number of EOFs used in SiFSAP to represent radiances and geophysical parameters.

| *Hyperspectral IR Instruments* | | *Number of Channels* | *Number of EOFs used* |
|---|---|---|---|
| AIRS | LWIR | 1262 | 50 |
| | MWIR | 602 | 35 |
| | SWIR | 514 | 40 |
| CrIS (NSR) | LWIR | 713 | 50 |
| | MWIR | 433 | 30 |
| | SWIR | 159 | 25 |
| CrIS (FSR) | LWIR | 713 | 50 |
| | MWIR | 863 | 40 |
| | SWIR | 865 | 30 |
| IASI | LWIR | 2260 | 50 |
| | MWIR | 3160 | 60 |
| | SWIR | 3041 | 80 |

| *Geophysical Parameters* | *Number of EOFs* |
|---|---|
| Temperature | 20 |
| Water Vapor | 15 |
| Carbon Dioxide | 1 |
| Ozone | 10 |
| Carbon Monoxide | 4 |
| IR Surface Emissivity | 8 |
| MW Surface Emissivity | 5 |




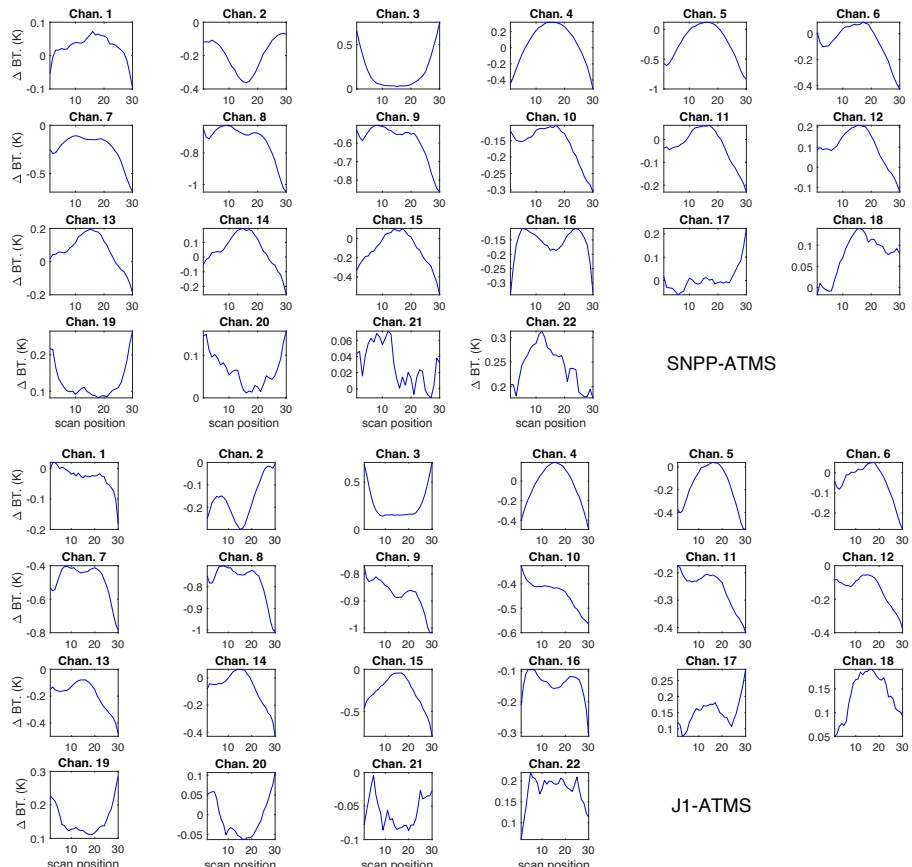

**Figure 8 Global mean bias (in BTs) predicted by the SiFSAP algorithm for ATMS measurements onboard of SNPP and NOAA20(JPSS-1) on April 30th, 2020.**



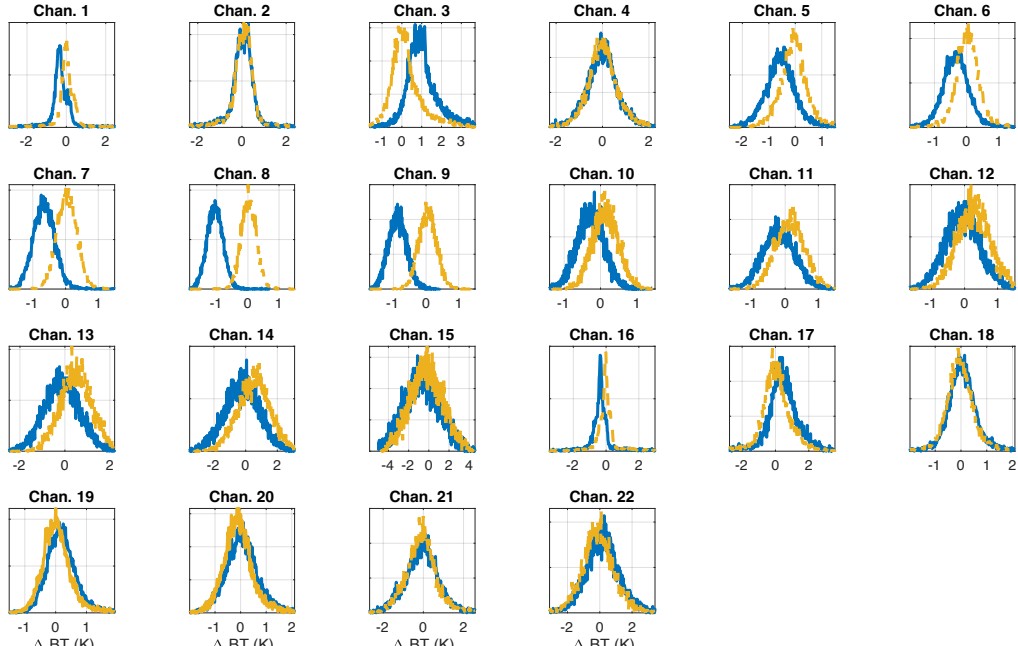

**Figure 9 Blue solid curves: histograms that illustrates the distribution of biases in different SNPP-ATMS channels for over the land measurements during April 30th, 2020 at the 52.725o scan position, being derived from the study discussed in Section 3.3**



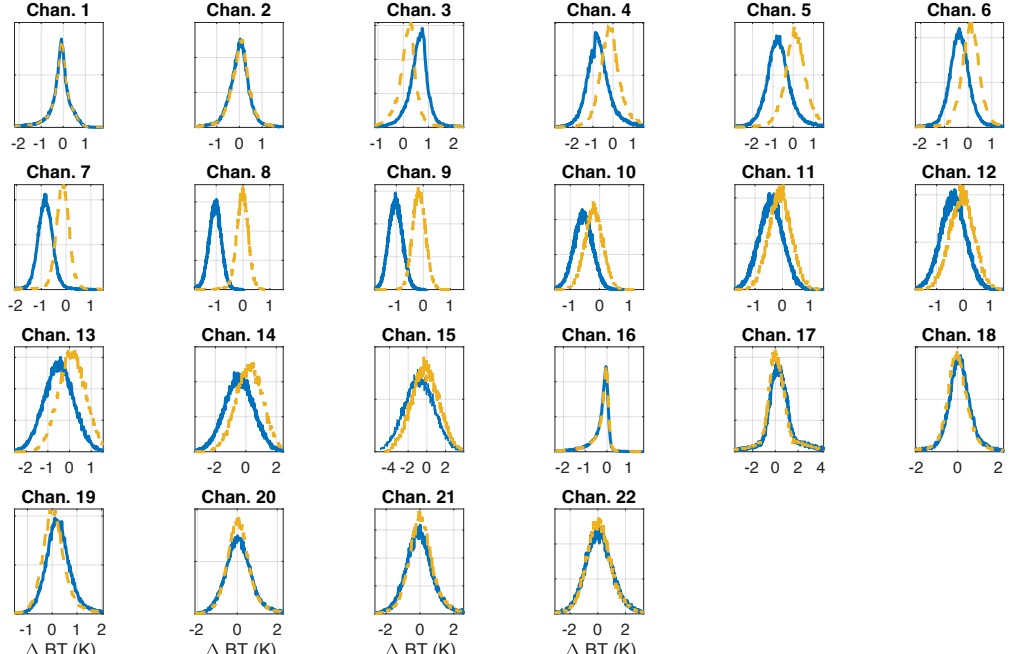

**Figure 10 Blue curves: similar to Figure 9 but for histograms that illustrate the distributions of biases in different NOAA20-ATMS channels for over the ocean measurements the 52.725° scan position. Yellow dash curves: histograms of biases after the correction following the regression-prediction scheme.**






**Figure 11 Left column: Spectral fitting error covariances (normalized by diagonal elements) used for SNPP-CrIS SiFSAP algorithm; Right column: Corresponding magnitude of the spectral fitting uncertainty for each CrIS channels (quantified as differential temperature at 280 K); A – For over ocean ascending observations; B – For over land ascending observations; C – For over ocean descending observations; D – For over land descending observations.**




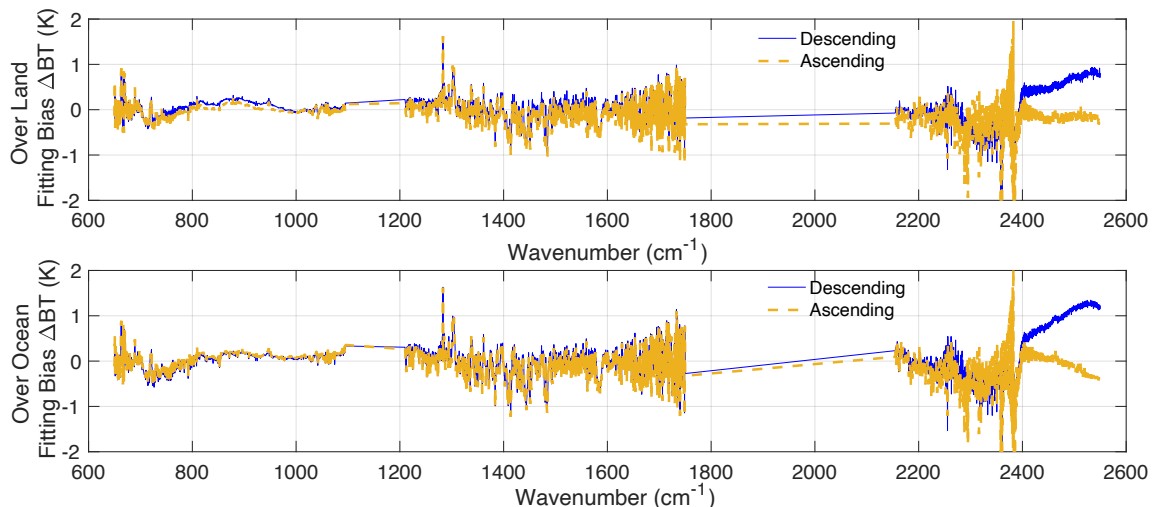

**Figure 12 Global scale daily mean spectral fitting bias achieved by SiFSAP for SNPP-CrIS observations on January 14, 2016.**

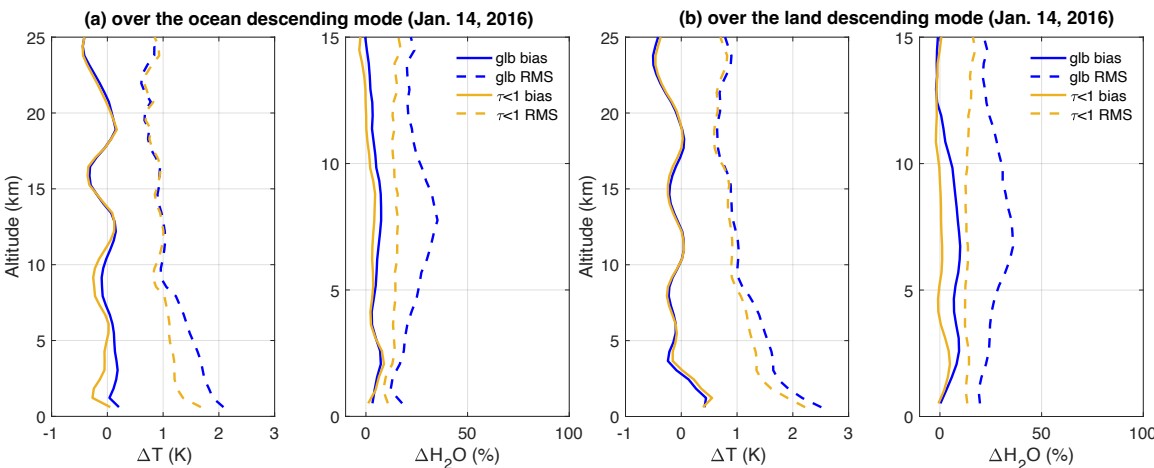

**Figure 13 Error statistics of global temperature and water vapor profiles retrieved from SNPP-CrIS/ATMS descending observations with respect to ECMWF for (a) over the ocean scenes and (b) over the land scenes. Solid lines: biases of temperature and water vapor profiles; dashed lines: RMS errors of the temperature and water vapor profiles. In addition to the bias and RMS values for all CrIS/ATMS descending measurements, the statistics for observations of under either clear sky or thin cloud (with cloud optical depth less than 1.0) are explicitly plotted to illustrate the impact of cloud on the retrieval accuracy of temperature and water vapor profiles at low altitudes.**





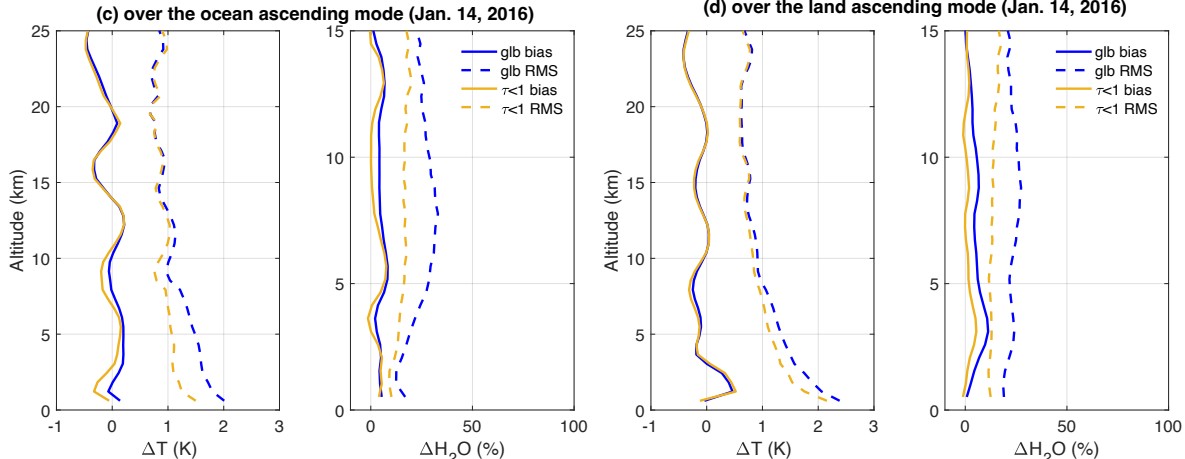

**Figure 14 Error statistics of global temperature and water vapor profiles retrieved from SNPP-CrIS/ATMS ascending observations with respect to ECMWF for (c) over the ocean scenes and (d) over the land scenes.**

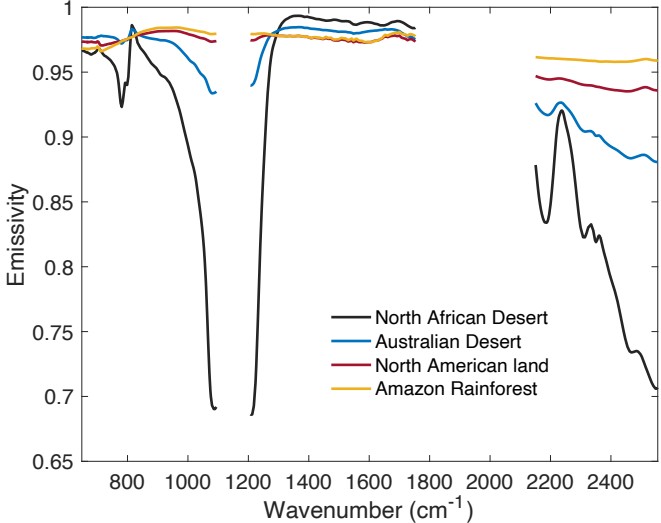

**Figure 15 Sample land emissivity spectra from SiFSAP of SNPP CrIS.**






**Figure 16 A1 and A2 – SiFSAP surface emissivity at 11 μm for January 14, 2016 and August 9, 2017, respectively; B1 and B2– MODIS surface emissivity at 11 μm for January, 2016 and August of 2017, respectively; C1 and C2 – MODIS monthly NDVI values for January, 20120016 and August of 2017, respectively; D1 and D2 – MODIS monthly snow coverage for January, 2016 and August of 2017, respectively; The area with relatively thicker clouds single (cloud optical depth larger than 1.0) are filtered out in subplots A1 and A2.**



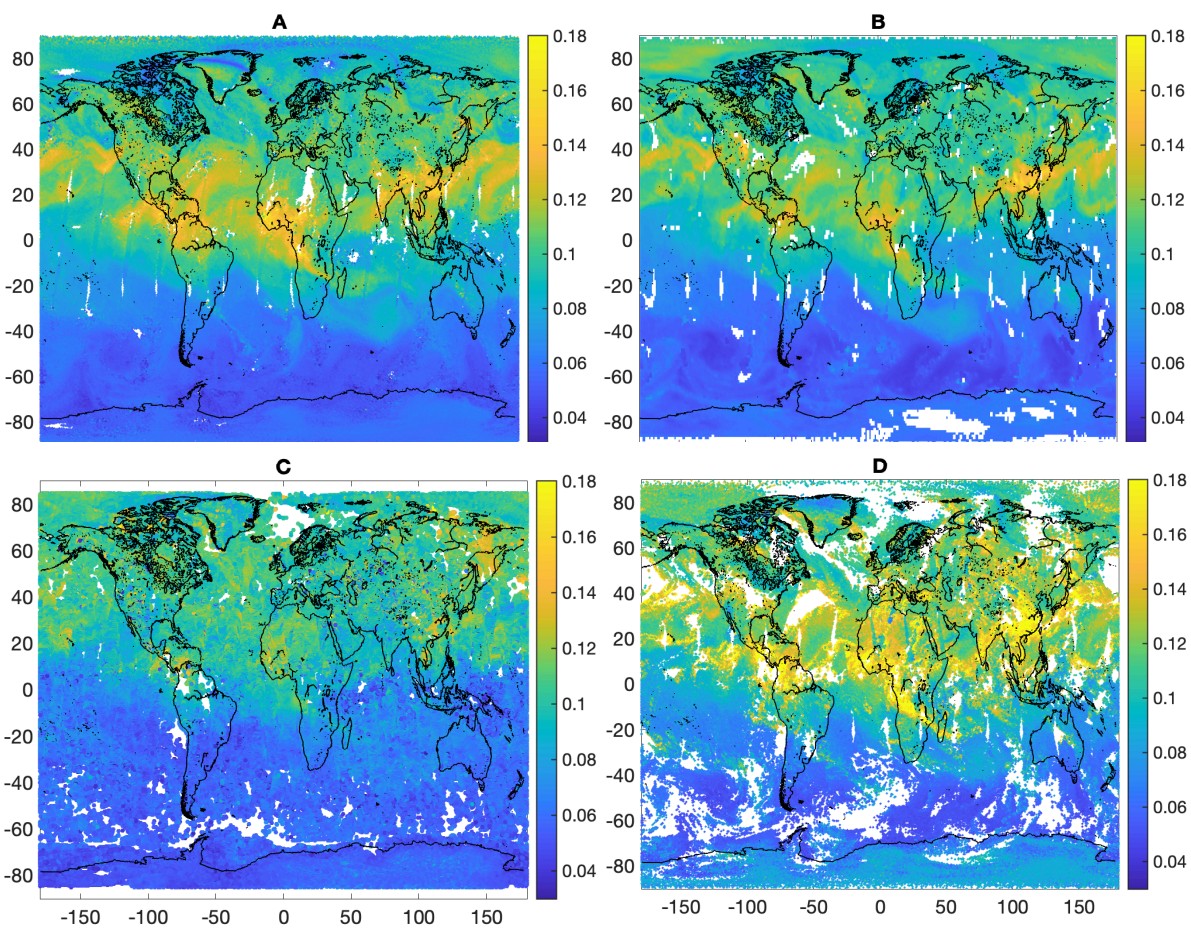

**Figure 17 A and B are map plots of VMR (ppmv) of CO at 500 hPa from SNPP CrIS SiFSAP and SNPP CrIS CLIMCAPS for May 12th, 2020, respectively; C shows the corresponding MOPITT CO VMR at 500hPa (during May 10-14, 2020, to ensure a global scale spatial coverage). Metop-B IASI FORLI CO VMR at 500 hPa is plotted in D.**





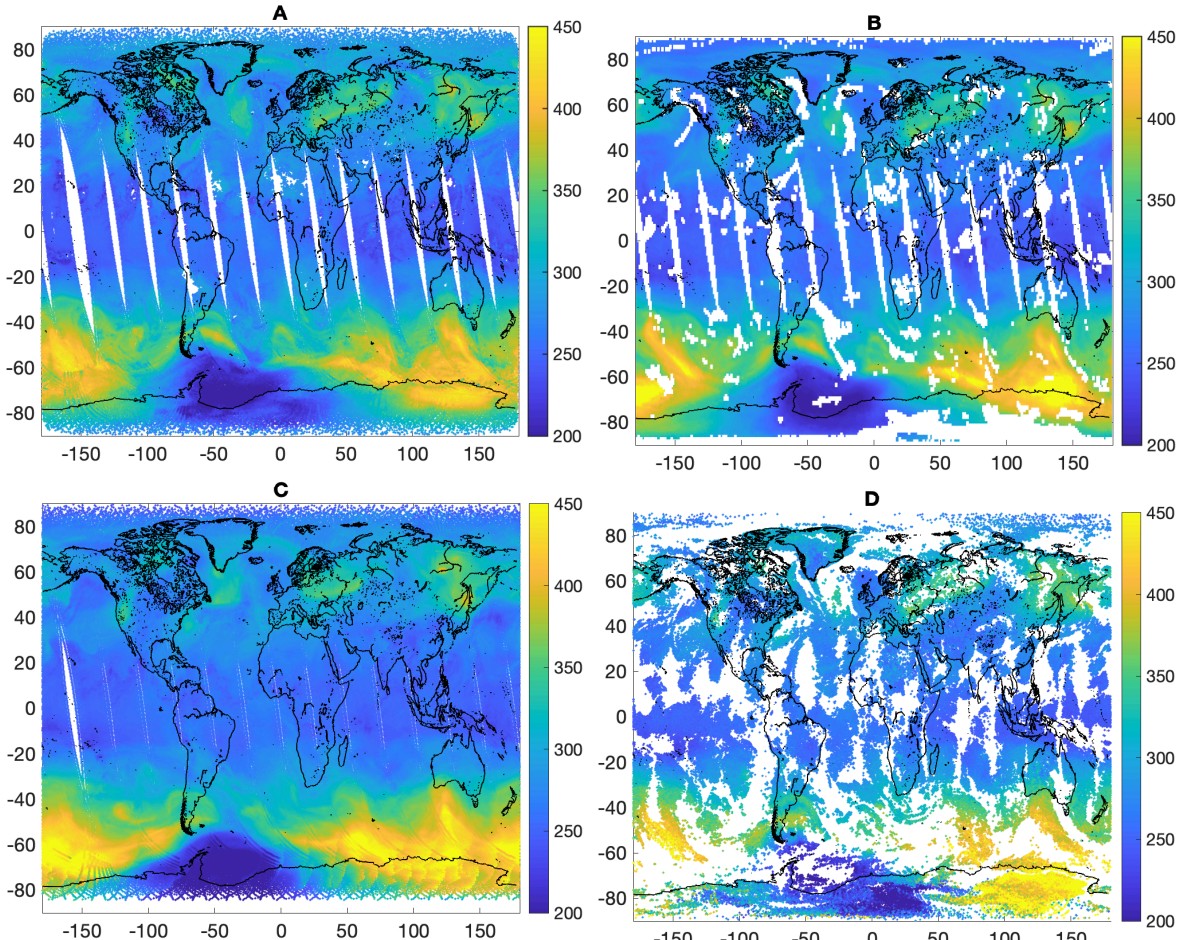

**Figure 18 O3 total column amount (DU) retrieved from satellite-based observations on September 20th, 2019 (A – SNPP/CrIS SiFSAP; B – SNPP/CrIS CLIMCAPS; C – SNNP-OMPS; D – Metop-B/IASI FORLI).**




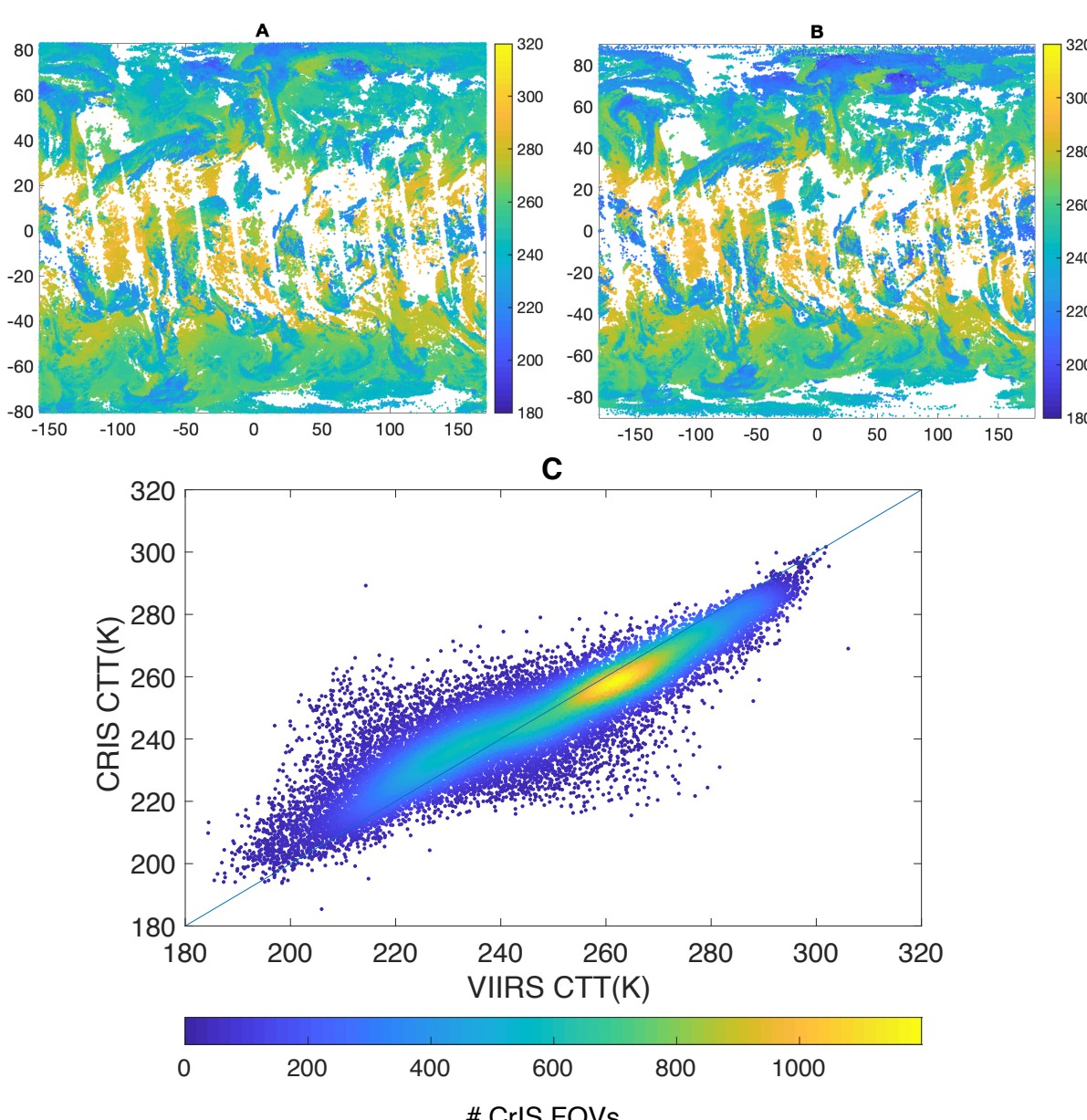

**Figure 19 Cloud top temperature (K) for January 1st, 2016 from SNPP/CrIS SiFSAP (A) and that from SNPP/VIIRS cloud data products collocated to CrIS footprints (B). C is the corresponding scatter plot.**


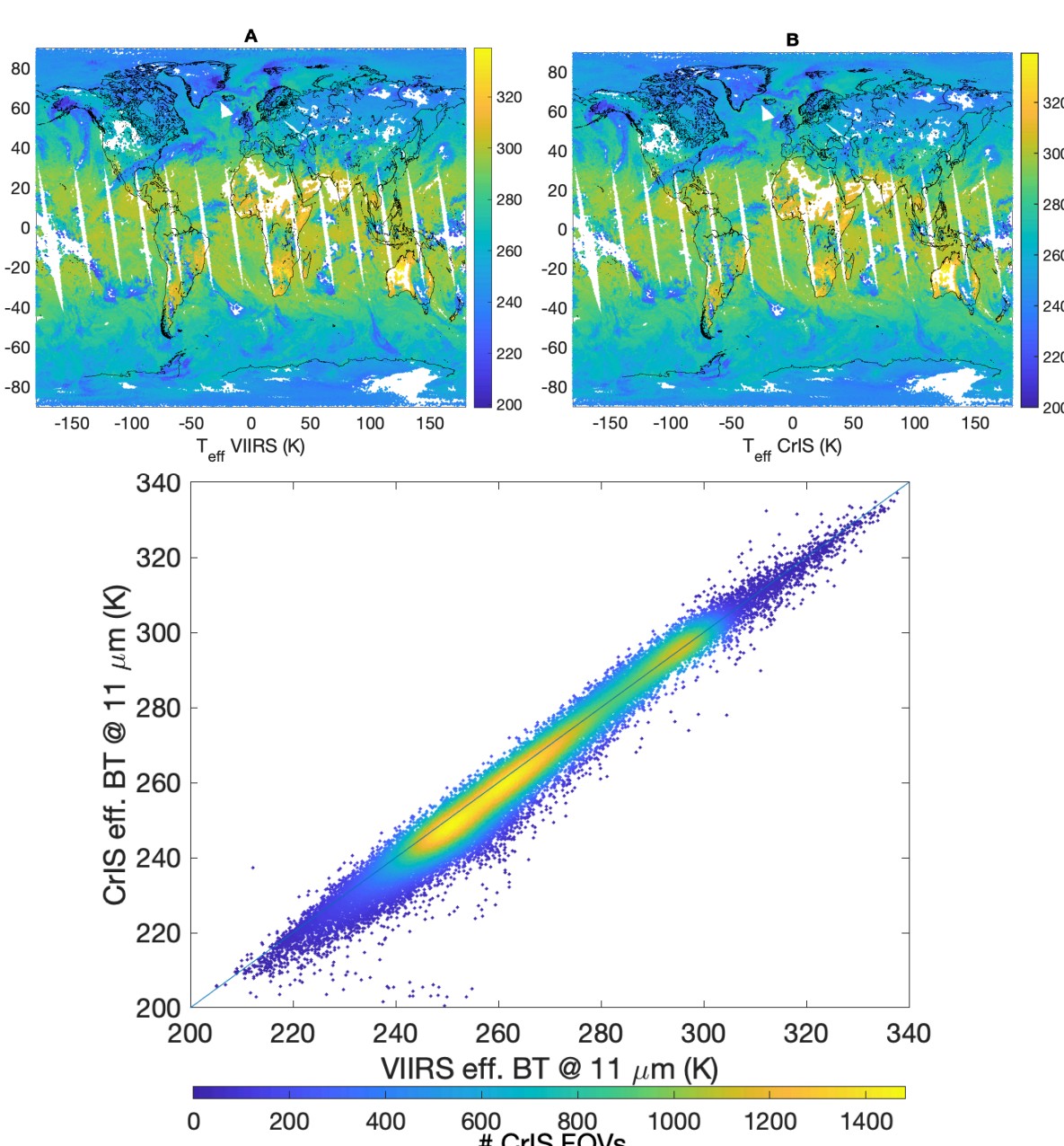

**Figure 20 Effective brightness temperature (K) for January 1st, 2016 from SNPP/CrIS SiFSAP (A) and that from SNPP/VIIRS cloud data products collocated to CrIS footprints (B). C is the corresponding scatter plot.**


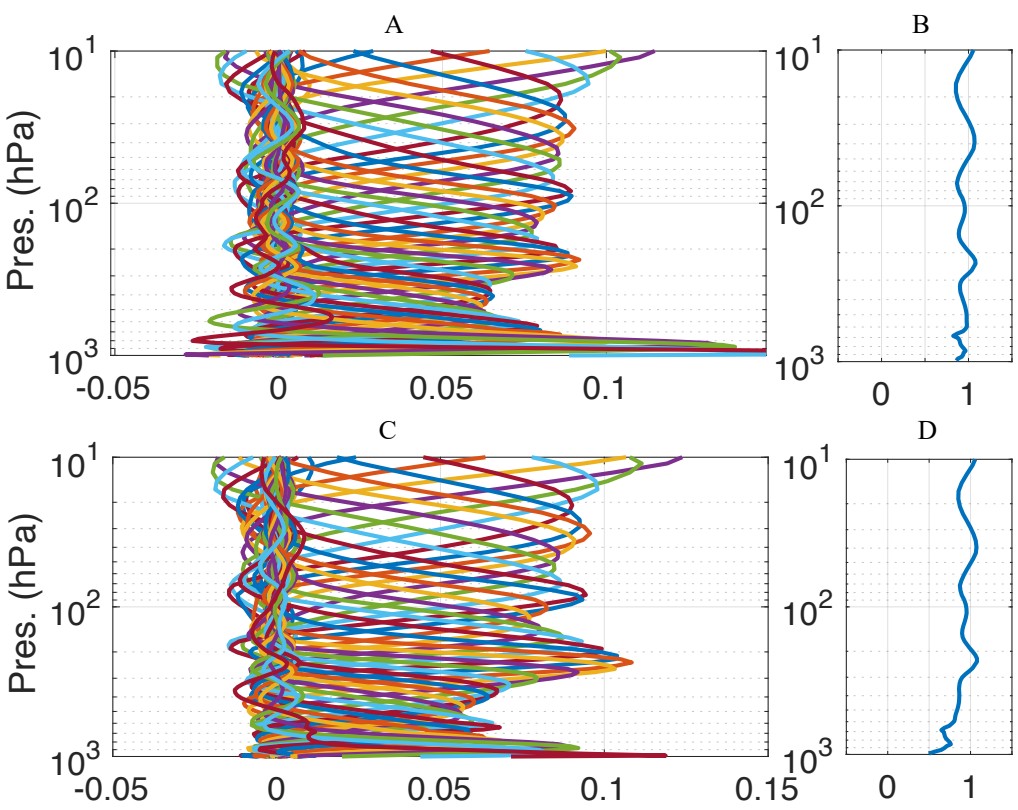

**Figure 21 Sample temperature averaging kernels from SNPP/CrIS SiFSAP;   A – averaging kernel under a clear sky condition; B – sum of averaging kernel rows at different pressure levels. C and D represent those under a cloudy sky condition.**





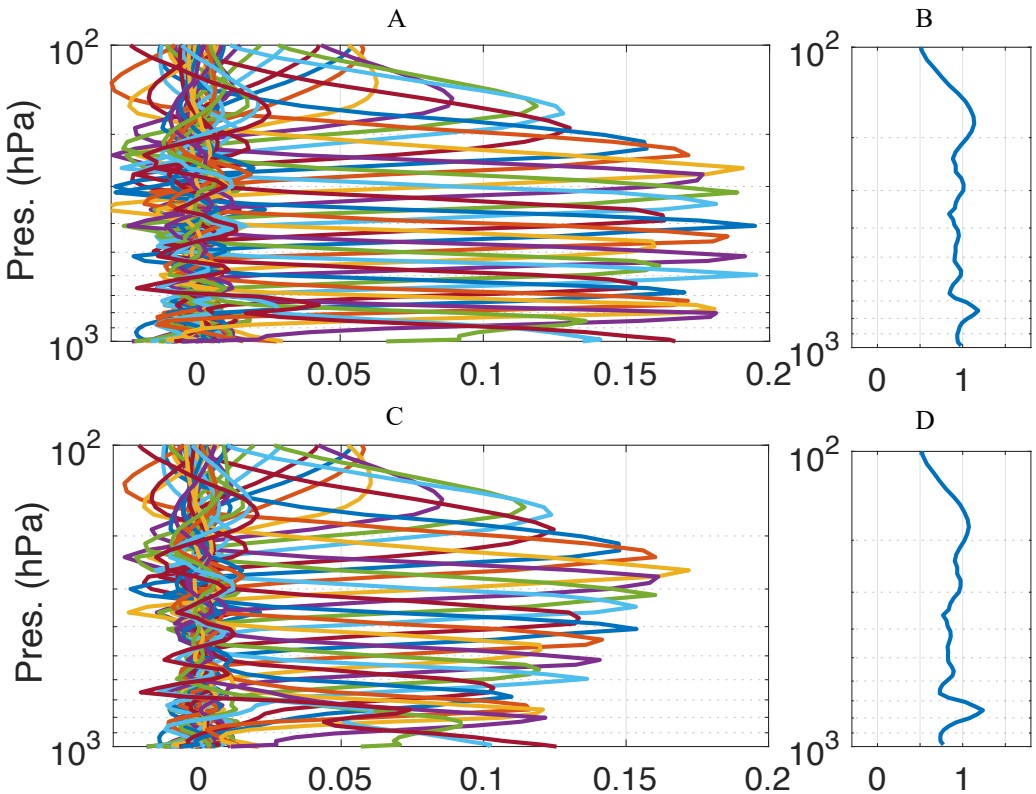

**Figure 22 Similar to Figure 21 but for water vapor retrievals.**





**Figure 23 Ozone averaging kernels of September 20th, 2019 from SiFSAP of SNPP CrIS for different latitudinal regions.**






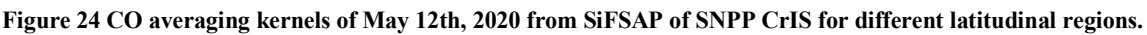

**Figure 24 CO averaging kernels of May 12th, 2020 from SiFSAP of SNPP CrIS for different latitudinal regions.**