# Peer review of "Single field-of-view sounder atmospheric product retrieval algorithm: establishing radiometric consistency for hyper-spectral sounder retrievals"

_EGUsphere, 2023_

## Author Response (AR1)

We really appreciate the time and effort by the reviewer. The suggestions and comments are indeed very helpful. We are thankful for the opportunity to revise our work and believe that the reviewers' feedbacks have truly improved the quality and rigor of our study.

We are delighted to submit the revised version of our manuscript in response to the reviewers' feedbacks. In this letter, we provide a point-by-point response to each reviewer's comments and describe the changes made in the revised manuscript.

**I.    Response to Reviewer 1**

Comment 1:

- **Section 1** (line 84-85): Why are the authors using lookup-tables for the modelling of the multiple scattering in clouds rather than the optical properties of water and ice directly? How were these look-up tables created. Which optical properties are the look-up tables based on? How much does the choice of optical properties affect the forward model?

Response :
PCRTM does not directly incorporate a multiple scattering scheme such as DISORT or adding-doubling since it will dramatically increase computational time and make analytical Jacobian calculations impractical. In PCRTM, cloud radiative transfer calculations are done using pre-computed cloud transmittance and reflectance. Effective cloud transmittance and reflectance are parameterized as a function of cloud optical depth, cloud particle size, and the satellite zenith angle. The cloud optical depth is referenced to a visible wavelength at 550 nm. The effective reflectance and transmittance have been calculated using DISORT. The single-scattering properties of the ice cloud were calculated using the method developed by Yang et al. (2013) and Baum et al. (2014) The optical properties of water cloud were calculated using the parameterization scheme developed by Hu and Stammes (1993). The impact of choice of optical properties on the forward model is not specifically characterized in this paper. However, it has been accommodated as part of the forward model error along with other error sources (e.g. spectroscopic error, errors due to the complicated vertical structure of clouds, etc.). The forward model error is addressed in the retrieval process via the bias correction and the fitting constraints (quantified by the spectral radiance error covariance).

References:
 P. Yang, L. Bi, B. A. Baum, K. Liou, G. W. Kattawar, M. I. Mishchenko, and B. Cole, "Spectrally consistent scattering, absorption, and polarization properties of atmospheric ice crystals at wavelengths from 0.2 to 100 μm," J. Atmos. Sci. **70**, 330–347 (2013).
 B. A. Baum, P. Yang, A. J. Heymsfield, A. Bansemer, B. H. Cole, A. Merrelli, C. Schmitt, and C. Wang, "Ice cloud single-scattering property models with the full phase matrix at wavelengths from 0.2 to 100 μm," J. Quant. Spectrosc. Raiat. Transfer **146**, 123–139 (2014).
 Y. Hu and K. Stamnes, "An accurate parameterization of the radiative properties of water clouds suitable for use in climate models," J. Climate **6**, 728–742 (1993).

Comment 2:

- **Section 2** (line 117): Why use a different forward model from PCRTM to compute the MW radiances. Could this not be done by PCRTM, thus negating the use of a second RT model and simplifying the overall algorithm.

  In Figure 2 would it be possible to plot both the mean error and its standard deviation.

Response :

Currently the PCRTM does not include the MW forward simulation module. We would like to develop a MW forward simulation module for PCRTM in the future.

Figure 2 shows the spectral fitting of PCRTM simulation to real AIRS, CrIS, IASI observations. We just show one spectral sample for each sensor, therefore, there is no statistical analysis applied here to derive mean and standard deviation values.

Comment 3:

  **Section 3.1:** These two sections on the Optimal Estimation process could be a little bit clearer. From my understanding the IR retrievals are done entirely in PC space, i.e., both the radiance vectors and the state vectors are projected onto their EOFs. However, for the MW computations only the state vector is projected into PC space. Is that correct?
  Since the radiances are projected into PC space, the instrument and model error covariance matrices need to be projected into PC space as well. How is that done? I assume that this is not a straightforward process and that there are "errors" involved in doing this. I would think, for example, that if the measurement errors for the hyperspectral channels are uncorrelated in physical space, then they will not necessarily be uncorrelated in PC space. Could the authors say a bit more about this or provide references.

  It might be worth mention in section 3.1 that the Levenberg–Marquardt method is used for the minimisation rather than just referring to two papers (Line 159). What is the difference between SR in equation 3 and Sε in equation 4. Is Sr just: Sε/a where a is a tuning parameter. It also states that provides the constraint in the radiance domain, which I find a bit confusing as I thought that all the retrieval is done in PC space.

Response :

You are right that both IR radiance vectors and geophysical state vectors are projected onto their EOFs. MW sounders have limited number of channels so that using EOFs to represent MW BTs is not necessary. For the first step MW-only retrieval, the same minimization scheme is used to find solution of geophysical variables in the EOF domain. Extra discussions about the dimension of radiance vector have been added in Section 3.1.

Principal Component Analysis (PCA) has been known as an effective mean of eliminating random noise. Therefore, instrument measurement random errors that are uncorrelated in physical space can be effectively filtered out after converting the spectral radiances into PC domain using limited number of EOFs. The PC filtered radiances are represented by EOFs accounting for most of the radiance variation and can therefore still precisely retain the spectral information content. The SiFSAP solution fits the PC filtered radiances instead of the radiances directly measured by the instrument. Therefore, instrument random noise introduced uncertainty does not need to be included in the error covariance matrix. The error covariance matrix only needs to account for uncertainty defined by the model error which are correlated in physical space. The uncorrelated instrument error will only become a concern if we choose to fit the radiances of selected channels instead of PCs, like other algorithms such as NUCAPS, CLIMCAPS, and AIRS v7.

The minimization method used in SiFSAP is not the traditional Levenberg–Marquardt method. It can be viewed as a modified Gauss-Newton approach. It works more effectively and efficiently than the traditional Levenberg–Marquardt method. Technical details of the method are not the focus of this paper, so we chose not to put a lot of discussions in the paper but encourage readers to find more details in the two reference papers. $S_\epsilon$ is the error covariance matrix that is derived based on the estimation for the forward model error. $S_\epsilon$ is also used to define the cost function of the final solution. $S_R$ is tuned as $S_\epsilon/a$ where $a$ changes for each minimization step (Wu et al. 2017). In this way, the step size of $x_n - x_{n-1}$ is tightly controlled at the beginning of the minimization procedure when $x_n$ is still in a non-linear region (far away from the final solution). $a$ is relatively large at the beginning so that the retrieval follows a gradient descent trajectory. As $a$ decreases after each iterative step, the weight of the measurement contributed information content increases so that the retrieval eventually approaches a Gaussian–Newton process.

The terminology 'radiance domain' is indeed a not well-defined term. It can be interpreted as 'spectral(channel) radiance domain' or 'PC radiance domain.' We have replaced 'radiance domain' in the text with 'measurement domain.' It refers to the vector of channel radiances for the MW only retrieval and the vector that combines the PC scores of IR radiance and the MW radiances (defined by Equation 6) for the IR+MW combined retrieval. Below is the text added/rearranged in the draft (lines 257- 268):

"The use of PC representation allows us to use all spectral channels of IR sensors and filter out instrument random noise. The solution $x$ includes all retrieved parameters that are used to quantify atmospheric vertical profiles, cloud information, and surface properties in the SiFSAP system. The dimension of the state vector $x$ also needs to be limited to reduce the computational cost and ensure the numerical stability. For example, atmospheric vertical profiles are usually not directly represented as level (or layer) quantities on a high vertical resolution pressure grid in a retrieval system. Retrieval algorithms including NUCAPS, CHART and CLIMCAPS use a linear

combination of pre-defined trapezoidal functions to represent vertical profiles. The principal component (PC) analysis is used to reduce the dimension of the geophysical state vector $x$ in SiFSAP. Atmospheric profiles and surface emissivity spectra are projected onto a set of pre-computed EOFs. Table 2 lists the dimension of measurement and geophysical state vectors used in SiFSAP. Both the IR+MW and MW only retrieval follow the same minimization scheme to find the solution of $x$ in the EOF domain. The dimensions of $r$, $S_r$, and $K$ are $1 \times N_{mw}$, $N_{mw} \times N_{mw}$, $N_{mw} \times N_x$ for the first-stage MW only retrieval, and $1 \times (N_{mw} + N_{ir})$, $(N_{mw} + N_{ir}) \times (N_{mw} + N_{ir})$, $(N_{mw} + N_{ir}) \times N_x$ for the second-stage IR+MW combined retrieval. Here $N_x$ is the length of the geophysical state vector $x$."

- **Section 3.2:** The section on how the ozone "a priori" is constructed is not very clear. Maybe it would be better to start with the datasets that are being used (MOZART, ECMWF and MERRA) and then describe step-by-step how the latitude-tropopause height stratification is achieved for the ozone a priori.

  Figure 4 is not labelled as (a), (b), (c) and (d) even so it is mentioned in the caption.

Thanks for the suggestion. We have made corresponding changes and rearranged the text in Section 3.2 to better describe the procedure of building and using the ozone *a priori*. Below is the updated text from line 391 to line 407:

"*A priori* for Ozone is generated using a synergistic dataset that combines data from the Model for Ozone And Related chemical Tracers (MOZART), ozone sonde measurements, the European Centre for Medium-Range Weather Forecasts (ECMWF) analysis, and the Modern-Era Retrospective analysis for Research and Applications (MERRA). The synergistic dataset includes more than 400,000 ozone profiles and collocated temperature profiles. Those profiles are globally distributed and provide adequate coverage for seasonal variabilities. The ozone and temperature profiles are binned into 18 10-degree latitudinal zones with each zonal group being further stratified into 13 tropopause-dependent sub-groups. The tropopause height values are derived as the lowest level at which the temperature lapse rate decreases to 2K/km or less. To further cover the seasonal variation characteristics of the ozone climatology, a linear regression relationship between the ozone profiles and the collocated temperature profiles are derived for each latitude-tropopause sub-group. The *a priori* covariance of each sub-group are derived as the regression-prediction uncertainty using the temperature and ozone data and saved as a static database. With a given tropopause height and a latitude, an individual retrieval is first assigned to a sub-group so that the *a priori* covariance used in the SiFSAP system can be directly loaded. The The first-guess values used for the ozone retrieval are obtained using the pre-established regression relationship of the assigned sub-group and the temperature profiles from the first-step MW retrieval. The SiFSAP system provides the option of using the tropopause height from either the real-time forecast data provided by National Centers for Environmental Prediction (NCEP) or that derived using temperature profiles from the first step MW retrieval. Both options are well suited for near-real-time applications."

The missing labels in Figure 4 have been added.

- **Section 3.3:** In figure 9 it does not say what the yellow line represents.

The missing description in the caption has been added in Figure 9.

- **Section 4.1:** Would it be possible to include the standard deviation or RMS in figure 12.

We can include the standard deviation or RMS in Figure 12. We chose not to plot the standard deviation or RMS along with the bias to better highlight the spectral feature of the systematic fitting residuals. The standard deviation of the fitting residuals generally follows the error covariance we estimated (shown in Figure 11), except in some spectral region where the magnitude of the instrument random noise is larger than the model error. We focus on the bias since daily mean spectral fitting bias is critical factor for the evaluation of using SiFSAP for climate studies. We will demonstrate in the future publications more analysis on the spectral fitting rms errors. Those errors are scene dependent and can give us more insights about the accuracy of the individual retrievals under different surface and cloud conditions.

- **Section 4.5:** Since clouds are made up of several layers, how does the scheme ensure that the optical properties of the single layer approximate those of clouds spread over several layers more or less accurately. Would it be possible to have cloud spread over several layers and if so, why is this approach not taken.

  The authors say that the global scale spatial distribution of the cloud top temperature from SiFSAP agrees well with that from the VIIRS cloud product except in the Arctic region. Why is that.

This is a very good question. Although we have demonstrated in the paper that a single layer cloud scheme can effectively approximate the radiative transfer contribution from multiple cloud layers for some cases (Wu et al. 2017). However, a simplified cloud model is indeed one of the error sources of the retrieval. Currently, such kind of error is mitigated by the bias correction and the accommodation for the spectral fitting uncertainty in the error covariance. We are working on improving the retrieval algorithm using a multiple layer cloud scheme. PCRTM can be used to do a multiple layer cloud radiative transfer simulation. However, the crosstalk issue due to the ill posed nature of the retrieval will be more complicated under a multiple layer cloud scheme. It is not straight forward to retrieve cloud properties of multiple layers following the current optimization scheme. We may need to rely more on *a prior* to retrieve multiple layer cloud properties. This is an on-going research work, and more results will be published in the future.

VIIRS is known for its challenge to retrieve semitransparent ice cloud due to the absence of infrared (IR) water vapor and $CO_2$ absorption channels. There were studies showing that the cloud mask over polar regions can be improved if VIIRS measurements can be supplemented with constructed IR water vapor and $CO_2$ absorption channel radiances using CrIS spectral

measurement in 6.7 µm and 15 µm region (Li et al., 2020). The publicly available VIIRS cloud data products used for the validation in the draft does not use those constructed channel radiances. Therefore, we believe the error in the VIIRS cloud data product can be the cause of the disagreement in the Arctic region.

Reference:
Li, Y., Baum, B. A., Heidinger, A. K., Menzel, W. P., and Weisz, E.: Improvement in cloud retrievals from VIIRS through the use of infrared absorption channels constructed from VIIRS+CrIS data fusion, Atmos. Meas. Tech., 13, 4035–4049, https://doi.org/10.5194/amt-13-4035-2020, 2020.

**Other**

Line 35                    : … has been widely recognised.

The error has been corrected.

Line 149                   : … the representation error.  A solution…

The error has been corrected.

Line 204                   : These profiles include European Centre for Medium-Range Weather Forecasting (ECMWF) reanalysis data, …

The error has been corrected.

Line 247:                  and values for given individual profiles are obtained by fitting the vertical profiles according to the function defined by equation (18).  Equation (18) refers to something very different.

Sorry, it should be equation (14). The error has been corrected.

Line 318                   : The abbreviation SNPP is never defined.

Both the full name of SNPP and JPSS have been added.

Line 375                   : … of hyperspectral sounder instruments. SiFSAP O3 data…

The error has been corrected.

**II.    Response to Reviewer 2**

Comment 1:

- When mentioning Dual-Regression (DR), I suggest to also cite Smith and Weisz, 2017 (http://dx.doi.org/10.1016/B978-0-12-409548-9.10394-X), which builds on the original DR algorithm by correcting for the vertical-resolution alias error. Although this advanced DR method is not an optimal estimation or a physical retrieval scheme in the traditional sense it can be considered as a fast-physical method since it incorporates radiative transfer model calculations (also from PCRTM) for every single FOV (however, only clear-sky PCRTM calculations are utilized to adjust the clear and cloudy regression retrievals).

Response:
   The suggested reference has been added.

Comment 2:

- Please consider mentioning imager plus sounder fusion work as well, which increases the sounder's spatial resolution even further from its native SFOV; for instance Smith et al., 2020 (https://doi.org/10.1175/JTECH-D-19-0158.1), which expands on the original SFOV Dual-Regression algorithm by combining multiple polar overpasses of IASI and CrIS with geostationary ABI data to improve not only the spatial resolution but also the temporal resolution of hyperspectral retrievals, extremely beneficial for data assimilation and now- and forecasting operations. Further, more recent work incorporates microwave data as well (initial results are shown at https://www.ssec.wisc.edu/hufusion/)

Response:
   The image plus sounder work is cited in the Introduction Section now.

Comment 3:

- To be consistent please use either 'radiative closure' or 'radiometric closure'.

Response:

   'radiometric closure' has been replaced with 'radiative closure.'

Comment 4:

- Line 58: The sentence "Existing SFOV products obtained using non-physical …" needs to be elaborated on. Although 'closure' is defined later in the introduction

section it should be alreday explained here what the authors mean by 'radiometric closure'.

Response:

We have rephrased and rearranged the text in the Introduction Section following the suggestion.

Below is the new text from line 58 to line 78:

"As compared with cloud-clearing based results, existing SFOV products (e.g. dual-regression and IASI PPF for cloudy-sky cases) are beneficial for data assimilation and now- and forecasting operations because of the higher sptial resolution. Smith (et al., 2020) have demonstrated that combining multiple polar overpasses of IASI and CrIS dual-regression retrieval with geostationary satellite Advanced Baseline Imager retrieval to improve not only the spatial resolution but also the temporal resolution of hyperspectral retrievals. Those retrieval schemes do not use optimal estimation based physical retrieval methodology and do not establish radiative closure by their nature. Establishing radiative closure, i.e. the radiometric consistency of the TOA spectra from radiative forward modelling using retrieved geophysical properties with respect to the observations, is critical to studies of climate trends and anomalies. The accuracy of climate trends derived from hyper-spectral IR observations depends on the radiometric accuracy of the measurements and a rigorously defined inverse relationship that links the measurements to the climate variables of interest (e.g. Liu et al., 2017). The closure in physical retrieval schemes including CHART, CLIMCAPS, NUCAPS and the hybrid IASI PPF can only be established for clear sky observations which just account for a small percentage of the global measurements. Without including cloud scattering in the forward simulations, the impact of radiometric uncertainty on the retrieved climate variables cannot be directly characterized. Estimation for radiometric errors and/or discontinuities and the corresponding impact on climate variables retrieved is critical for the construction of long-term climate anomalies and/or trends data record. From this perspective, a physical retrieval algorithm that establishes radiative closure by simulating cloud scattering in the radiative transfer process is more suitable to produce accurate, long-term climate data records."

Comment 5:

- Line 80: In "The SiFSAP retrieval algorithm has been developed ...." consider replacing "by improving the spatial resolution" to "by sustaining the hyperspectral sounder's spatial resolution" to avoid confusion

Response:

Suggested change has been made in the draft.

Comment 6:

- Line 89: Consider adding one of C. D. Rodgers' publications, e.g., Rodgers, 1976, when mentioning optimal estimation.

Response:

In Section 3.1, we cited Rodgers's book about optimal estimation. However, the reference is missing in the reference list. The error is fixed now.

Comment 7:

- Line 94: The sentence "Using stringent a priori reduces …." - especially the 'given faulty a priori' part - sounds awkward; please rewrite this sentence.

Response:

The sentence has been written as 'Using stringent *a priori* reduces the uncertainty in individual retrievals but can make the results more prone to systematic errors if *a priori* is not properly established.'

Comment 8:
Section 4.1, line 340: Consider citing the N. Smith. et al, 2015 (https://doi.org/10.1175/JAMC-D-14-0299.1) publication, which also discusses climate trends from all three hyperspectral sounders.

Response:
What we want to highlight here is that the accuracy of achieving radiative closure is important to the climate trend study. We need to assess spectral radiance fitting to ensure the capability of providing radiance 'closure'. Although N. Smith studies the methodology of deriving climate trends from all three hyper-spectral sounders, the radiance fitting assessment is irrelevant in her work. Citing this work may not be very helpful to both Smith's work and this paper here.

Comment 9:
Figures 4-7, 13, 14: If possible, please display the standard deviations as well.

Thanks for the suggestion. We had been back and forth with the option of plotting standard deviations or RMS values because different people have different preference. Displaying both standard deviations and RMS values in those Figures can make some part of the curves overlapping with each other and become less illustrative to readers. At this moment, we think it is better for us keep those Figures as they are. Really appreciate your understanding about this. We can send standard deviation plots to readers of this paper based on their request.

Technical comment :

- Line 35: Correct year of Lieu et al. is 2017 (not 2018)

Response:
It is actually Liu et al. 2018.

Liu, R., Su, H., Liou, K.-N., Jiang, J. H., Gu, Y., Liu, S. C. & Shiu, C.-J.: An assessment of tropospheric water vapor feedback using radiative kernels, Journal of Geophysical Research: Atmospheres, 123, 1499-1509, https://doi.org/10. 1002/2017JD027512, 2018.

- Line 63: Correct year of the Cousins and Smith paper is 1997 (not 1999).

Response: The error is now corrected.

- Line 120: The correct publication year of the CRTM v2.4. user's guide should be stated as 2020 (not 2005).

Response: The error is now corrected.

- Sect. 3.1: Please consider using lower case for vectors (e.g., state vector x) and upper case for matrices (e.g., averaging kernel matrix A) in all the equations.

Response: Suggested has been followed.

- Line 149: Remove 'and et al.'

Response: 'and et al.' is removed now.

- Line 184: Use 'scalar' instead of 'scaler'

Response: The misspelled word is corrected now.

- Line 204: I think the authors meant to cite Wu et al., 2017 (instead of saying Wan et al., 2017).

Response: The error is corrected now.

- Line 376: Please correct to 2022B for the second Xiong et al. reference.

Response: The error is corrected now.

- Line 409: Fu et al., 2018 is not included in the list of references.

Response: The missing refence is added now.

- Line 414: Boynard et al., 2018 is not included in the list of references.

Response: The missing refence is added now.

- In References: the correct year for Elsaesser et al. is 2019 (not 2020);

Response: The error is corrected now.

- and the reference for Liu, R., Su, H., ... 2018 was not mentioned in the text.

It is cited as Liu et al. 2018 in line 35.